

# Proposed standardized definitions for vertical resolution and uncertainty in the NDACC lidar ozone and temperature algorithms. Part 3: Temperature uncertainty budget

Thierry Leblanc [1], Robert J. Sica [2], J. Anne E. van Gijsel [3], Alexander Haefele [4], Guillaume Payen [5], and Gianluigi Liberti [6]

[1] Jet Propulsion Laboratory, California Institute of Technology, Wrightwood, CA 92397, USA
[2] Department of Physics and Astronomy, The University of Western Ontario, London, Canada
[3] Royal Netherlands Meteorological Institute (KNMI), Netherlands
[4] Meteoswiss, Payerne, Switzerland
[5] Observatoire des Sciences de l'Univers de La Réunion, CNRS and Université de la Réunion (UMS3365), Saint Denis de la Réunion, France
[6] ISAC-CNR, Via Fosso del Cavaliere 100, I-00133, Rome, Italy

*Correspondence to*: Thierry Leblanc (thierry.leblanc@jpl.nasa.gov)

Abstract. A standardized approach for the definition, propagation and reporting of uncertainty in the temperature lidar data products contributing to the Network for the Detection for Atmospheric Composition Change (NDACC) database is proposed. One important aspect of the proposed approach is the ability to propagate all independent uncertainty components in parallel through the data processing chain. The individual uncertainty components are then combined together at the very last stage of processing to form the temperature combined standard uncertainty.

The identified individual uncertainty components comprise signal detection uncertainty, uncertainty due to saturation correction, background noise extraction, the merging of multiple channels, the absorption cross-sections of ozone and $NO_2$, the molecular extinction cross-sections, the a priori use of ancillary air, ozone, and $NO_2$ number density, the a priori use of ancillary temperature to tie-on the top of the profile, the acceleration of gravity, and the molecular mass of air. The expression of the individual uncertainty components and their step-by-step propagation through the temperature data processing chain are thoroughly estimated. All sources of uncertainty except detection noise imply correlated terms in the vertical dimension, which means that covariance terms must be taken into account when vertical filtering is applied and when temperature is integrated form the top of the profile. Quantitatively, the uncertainty budget is presented in a generic form (i.e., as a function of instrument performance and wavelength), so that any NDACC temperature lidar investigator can easily estimate the expected impact of individual uncertainty components in the case of their own instrument. An example of a full uncertainty budget obtained from actual measurements by the JPL lidar at the Mauna Loa Observatory is also provided.

## 1 Introduction

The present article is the last of three companion papers that provide a comprehensive description of recent recommendations made to the Network for Detection of Stratospheric Change (NDACC) lidar community for the standardization of vertical resolution and uncertainty in the NDACC lidar data processing algorithms. More than 20 lidar instruments contribute long-term measurements to NDACC, as well as to the validation of satellite or aircraft



measurements. A wide range of methodologies and technologies is used for NDACC lidar instrumentation, which inherently raises the issue of consistency across the network, especially when using the lidar data to detect long-term trends, to perform intercomparisons, model or instrument validation, or when trying to ingest the data in assimilation models.

No comprehensive effort has been made until recently to facilitate a standardization of the definitions and approaches used in the NDACC lidar data processing algorithms. In 2011, an *International Space Science Institute* (ISSI) International Team of Experts ([http://www.issibern.ch/aboutissi/mission.html](http://www.issibern.ch/aboutissi/mission.html)) (henceforth "ISSI Team") was formed with the objective to provide recommendations on the use of standardized definitions or approaches for vertical resolution and the treatment of uncertainty in the NDACC lidar retrievals (**Leblanc et al., 2016a**). Our first

companion paper ("Part 1") **[Leblanc et al., 2016b]** reviews the recommendations made by the ISSI Team for the use of standardized definitions of vertical resolution. Our second companion paper ("Part 2") (**Leblanc et al., 2016c**) reviews the definitions and approaches proposed by the ISSI Team for a standardized treatment of uncertainty in the ozone DIAL retrievals. The present paper ("Part 3") presents a work similar to that presented in our "Part 2", but for the temperature lidar retrievals. The approach and recommendations described here apply to the

density integration technique (Hauchecorne and Chanin, 1980), but not to the Optimal Estimation Method (OEM) (Sica and Haefele, 2015) for which vertical resolution and uncertainties are computed implicitly by the OEM. Some concepts described here and in our "Part 2" companion paper may be used for the rotational Raman technique, but will not be discussed here. In the rest of this work, for brevity, every mention of "temperature lidar" will only refer to the retrieval of temperature using the density integration technique.

Middle atmospheric temperature profiles (15-80 km) have been measured by lidar for decades now using the density integration technique (e.g., Hauchecorne and Chanin, 1980; Keckhut et al., 1993; Keckhut et al., 2011). The corresponding temperature uncertainty budgets, as reported in the literature, have typically included statistical noise (e.g., Hauchecorne and Chanin, 1980), and less frequently other components such as saturation (pulse pile-up) (e.g., Leblanc et al., 1998), ozone absorption correction (Sica et al., 2001) or temperature initialization (Argall, 2007).

Using synthetic lidar signals, Leblanc et al. (1998) provided a review of the most common error sources made in the lidar temperature retrievals. Inter-comparison campaigns set up in the frame of NDACC have also contributed to assess lidar measurement uncertainties (Keckhut et al., 2004).

In this paper, we propose a standardized and consistent approach for the introduction and propagation of the uncertainty components contributing to the full temperature uncertainty budget. Reference definitions on uncertainty

are briefly reviewed in **section 2**. Based on these definitions, a standardized measurement model for temperature lidars using the density integration technique is proposed in **section 3**. Using this model, a complete formulation for the propagation of uncertainty through the temperature lidar algorithm is provided in **section 4**. An example of an actual temperature uncertainty budget is then provided in **section 5**, followed by a brief summary and conclusion. The structure of the present paper and the fundamentals described in it are very similar to those presented in our

"Part 2" companion paper (**Leblanc et al., 2016c**), and therefore the readers will find frequent references to this companion paper, which provides more details on many aspects reviewed here. Ultimately, the reader should refer to the ISSI Team Report (**Leblanc et al., 2016a**) for more details on all aspects covered in the present article.





## 2 Reference definitions

Two metrological concepts, namely *measurement model* and *combined standard uncertainty*, should be quickly introduced prior to propose a standardized approach for the treatment of the temperature uncertainty for the NDACC lidars. It is strongly advised to refer to our "Part 2" companion paper (Leblanc et al., 2016b), where these concepts

are discussed in more details, with key references to the metrological standards of the Bureau International des Poids et Mesures (BIPM) (JCGM 100, 2008; JCGM 200, 2008; 2012). Here we only provide a very brief overview.

### 2.1 Measurement model

For complex measurement techniques such as lidar, the retrieved temperature profile depends on multiple instrumental and physical parameters (see **section 3.2** thereafter). We therefore introduce the concept of

measurement model, which is a "*mathematical relation among all quantities known to be involved in a measurement*" (VIM art. 2.48 (JCGM 200, 2012)). Those quantities are referred to as the "input quantities", and the quantity derived from them is referred to as the "output quantity". A measurement model represents the mathematical architecture around which a standardized uncertainty budget can be built. The individual values $y$ of an output quantity $Y$ describing a measurement model that comprises multiple input quantities $x_n$ can be

approximated to the first order of its Taylor-expanded form:

$$y = f(x_1, x_2, ..., x_N) = y_0 + \sum_{n=1}^{N} \frac{\partial y}{\partial x_n} x_n \qquad (1)$$

The fully-expanded form of this equation is provided in our "Part 2" companion paper. Equation **(1)** is at the origin of the so-called *law of propagation of uncertainty* defined in the next paragraph.

### 2.2 Combined standard uncertainty

The definition of uncertainty recommended by the ISSI Team for use by all NDACC lidar measurements is the *combined standard uncertainty*. Standard uncertainty is defined in article 2.30 of the VIM (JCGM 200, 2012) as "*the measurement uncertainty expressed as a standard deviation*". The true values of a model's input quantities $x_n$ are unknown, and can be assigned a standard uncertainty $u_n$ characterizing their probability distribution. The output quantity's combined standard uncertainty $u_y$ is the "*standard measurement uncertainty that is obtained using the*

*individual standard uncertainties associated with the input quantities*". The input quantities' uncertainty components can either be estimated by "Type A" or "Type B" evaluations. A "Type A" standard uncertainty is obtained from a probability density function derived from an observed frequency distribution, while a "Type B" standard uncertainty is obtained from an assumed probability density function based on best available knowledge. When two input quantities $x_n$ and $x_m$ are correlated (i.e., their correlation coefficient $r_{nm}$ is not equal to zero), their

covariance must be taken into account. The combined standard uncertainty is equal to the positive square root of the combined variance obtained from all variance and covariance components using the "law of propagation of uncertainty" (art. 5.2 of the GUM (JCGM 100, 2008)) which, when using the notations of **Eq. (1)**, can be written:





$$u_y = \sqrt{\sum_{n=1}^{N}\sum_{m=1}^{N}\frac{\partial y}{\partial x_n}\frac{\partial y}{\partial x_m}\mathrm{cov}(x_n,x_m)} = \sqrt{\sum_{n=1}^{N}\left(\frac{\partial y}{\partial x_n}\right)^2 u_n^2 + 2\sum_{m=1}^{N-1}\sum_{n=m+1}^{N}\frac{\partial y}{\partial x_n}\frac{\partial y}{\partial x_m}r_{nm}u_n u_m} \qquad (2)$$

Equations **(1)-(2)** as well as other expressions described in **section 2** of our "Part 2" companion paper fully characterize a measurement model and the output quantity's combined standard uncertainty.

### 3 Proposed measurement model for the NDACC temperature lidars

In this section, a standardized lidar measurement model for the retrieval of temperature using the density integration technique is constructed. The approach implies the replacement of a single, complex temperature measurement model by the successive application of multiple, simpler measurement sub-models, which typically are specific transformations of the raw lidar signals. For each signal transformation, standard uncertainty can be evaluated in parallel for each independent uncertainty source. During the final data processing stage, all independent components

are combined together to obtain the temperature combined standard uncertainty.

### 3.1 Lidar Equation

As in most lidar applications, the fundamental equation at the source of the middle atmospheric temperature lidar retrieval using the density integration technique is the *Lidar Equation* (e.g., Hinkley, 1976). The equation describes the emission of light by a laser source, its backscatter at altitude $z$, its extinction and scattering along the laser beam

path up and back, and its collection on a detector. One form of the lidar equation is:

$$P(z,\lambda_1,\lambda_2) = P_L(\lambda_1)\frac{\eta(z,\lambda_2)\delta z}{(z-z_L)^2}\tau_{UP}(z,\lambda_1)\beta(z,\lambda_1,\lambda_2)\tau_{DOWN}(z,\lambda_2) \qquad (3)$$

$\lambda_1$ is the laser emission wavelength and $\lambda_2$ is the receiver detection wavelength

$P$ is the total number of photons collected at wavelength $\lambda_2$ on the lidar detector surface

$\delta z$ is the thickness of the backscattering layer sounded during the time interval $\delta t$ ($\delta z = c\delta t/2$, where c is the speed of

light)

$P_L$ is the number of photons emitted at the emission wavelength $\lambda_1$

$\eta$ is the optical efficiency of the receiving channel, including optical and spectral transmittance and geometric obstruction

$z$ is the altitude of the backscattering layer

$z_L$ is the altitude of the lidar (laser and receiver assumed to be at the same altitude)

$\beta$ is the total backscatter coefficient (including particulate and molecular backscatter)

$\tau_{UP}$ is the optical thickness integrated along the outgoing beam path between the lidar and the scattering altitude $z$, and is defined as:

$$\tau_{UP}(z) = \exp\left[-\int_{z_L}^{z}\left(\sigma_M(\lambda_1)N_a(z') + \alpha_P(z',\lambda_1) + \sum_i \sigma_i(z',\lambda_1)N_i(z')\right)dz'\right] \qquad (4)$$



$\tau_{DOWN}$ is the optical thickness integrated along the returning beam path between the scattering altitude $z$ and the lidar receiver, and is defined as:

$$\tau_{DOWN}(z) = \exp\left[-\int_{z_L}^{z}\left(\sigma_M(\lambda_2)N_a(z') + \alpha_P(z',\lambda_2) + \sum_i \sigma_i(z',\lambda_2)N_i(z')\right)dz'\right] \tag{5}$$

$\sigma_M$ is the molecular extinction cross-section due to Rayleigh scattering (Strutt, 1899) (thereafter called "Rayleigh cross-section" for brevity), $N_a$ is the air number density, $\alpha_P$ is the particulate extinction coefficient, $\sigma_i$ is the absorption cross-section of absorbing constituent $i$, and $N_i$ is the number density of absorbing constituent $i$. For altitudes between the ground and 90 km, the Rayleigh cross-sections can be considered constant with altitude, and therefore depend only on wavelength. The absorption cross-sections however are in most cases temperature-dependent, and should be taken as a function of both altitude and wavelength. Temperature is retrieved by inverting **Eq. (3)** with respect to the backscatter term $\beta$.

### 3.2 Inversion of the lidar equation for temperature retrieval

In the absence of particulate backscatter, the backscatter coefficient $\beta$, and therefore the lidar signal collected on the detector, is proportional to the air number density. Temperature is then calculated by vertically integrating air number density assuming hydrostatic balance and assuming that the air is an ideal gas (Hauchecorne and Chanin, 1980). This inversion technique works for both elastic scattering (Rayleigh backscatter by the air molecules) and inelastic scattering (normally, using vibrational Raman backscatter by the nitrogen molecules) (Strauch et al., 1971; Gross et al., 1997). For either technique, we can write a generic form of the backscatter coefficient as a function of air number density $N_a$:

$$\beta(z) = \sigma_\beta N_a(z) \tag{6}$$

For Rayleigh backscatter, the effective cross-section $\sigma_\beta$ is the molecular (Rayleigh) scattering cross-section at the emission wavelength $\lambda_1$:

$$\sigma_\beta = \sigma_M(\lambda_1) \tag{7}$$

For Raman backscatter, the effective cross-section $\sigma_\beta$ is the vibrational Raman scattering cross-section of a well-mixed gas (typically nitrogen) at the Raman-shifted wavelength $\lambda_2$, multiplied by the mixing ratio of the well-mixed gas (e.g., 0.781 for nitrogen):

$$\sigma_\beta = 0.781\sigma_{N2}(\lambda_1,\lambda_2) \tag{8}$$

Substituting into the lidar equation **Eq. (3)**, we obtain an expression of air number density as a function of the backscatter lidar signal:

$$N_a(z) = \frac{P(z,\lambda_1,\lambda_2)(z-z_L)^2}{\sigma_\beta \eta(z,\lambda_1,\lambda_2)\delta z P_L(\lambda_1)\tau_{UP}(z,\lambda_1)\tau_{DOWN}(z,\lambda_2)} \tag{9}$$

A temperature profile is then calculated assuming hydrostatic balance and assuming that the air is an ideal gas with a constant mean molecular mass:



$$T(z - \delta z) = \frac{N_a(z)}{N_a(z - \delta z)} T(z) + \frac{M_a}{R_a N_a(z - \delta z)} \overline{N_a}(z)\overline{g}(z)\delta z \qquad (10)$$

$T$ is the retrieved temperature, $M_a$ is the molecular mass of dry air, $R_a$ is the ideal gas constant, and $g$ is the acceleration of gravity. The horizontal bar above $N_a$ and $g$ represents the average value of $N_a$ and $g$ between $z$ and $z$-$\delta z$. An essential aspect of the method is that all altitude-independent terms (e.g., Rayleigh cross-section, lidar receiver efficiency) cancel out when computing the ratio of air number density at altitudes $z$ and $z$-$\delta z$.

A list of the most commonly used wavelengths is compiled in **Table 1**.

**3.3 Actual temperature measurement model proposed for standardized use within NDACC**

The actual temperature measurement model proposed for a standardized NDACC lidar temperature uncertainty budget is a real-world version of the theoretical model described in the previous paragraph after considering the technical limitations owed to the design, setup, and operation of a real lidar instrument.

First, several assumptions on the property of the atmosphere must be made to help reduce the complexity of our proposed measurement model. Specifically, uncertainty components associated with particulate extinction and backscatter will not be considered here. For Rayleigh backscatter channels, the bottom of the retrieved temperature profile is typically at 25-30 km where the atmosphere is normally "clean". Particulate matter contribution may occasionally be significant below 35 km in the presence of heavy stratospheric volcanic loading (e.g., Mount Pinatubo eruption in 1991). When present, the amount and physical properties of the particulate matter can be highly variable from site to site and from time to time, and very difficult to estimate. The standardized treatment of these uncertainty components is therefore too complex to be included in the present work. However it should be considered in a dedicated study using leverage from past work, for example work performed within the EARLINET project (D'Amico et al., 2015; Mattis et AL., 2016).

Secondly, the number of photons collected on the lidar detectors $P$, as it appears in **Eq. (9)**, is different from the actual raw lidar signals recorded in the data files. Signal corrections and numerical transformations related to the instrumentation are necessary. The backscattered signal is indeed altered by sky and electronic background noise, efficiency loss, signal saturation (pulse pile-up), and sometimes other non-linear effects that must be taken into account. Because of the wide range of lidar instrumentation, providing a unique expression for the parameterization of these effects is very challenging. Here we consider a few special cases representing the largest fraction of currently-operated NDACC lidar systems.

In order to transition from a theoretical to a real temperature measurement model, the following assumptions and transformations will be made:

1) For each lidar receiver channel, the actual raw signal $R$ recorded in the data files is represented by a vector of discretized values rather than a continuous function of altitude range:

$z \to z(k)$ and $R(z) \to R(k)$      for $k = 1, nk$

2) Only channels operating in photon-counting mode are considered in this measurement model. The estimation of the uncertainty due to analog-to-digital signal conversion is highly instrument-dependent, and therefore no



meaningful standardized recommendations can be made. However, an example of the treatment of the analog-to-detection uncertainty is provided in the ISSI team Report **(Leblanc et al., 2016a)**.

3) For each lidar receiver channel, the actual raw signal recorded the data files comprises altitude-dependent signal resulting from the laser light backscattered in the atmosphere, a constant (typically small) noise coming from the sky background light, and time-dependent (typically small) noise generated within the electronics (dark current and signal-induced noise). The noise components can be parametrized by either a constant, linear or non-linear function of altitude range.

4) In photon-counting mode, signals of large magnitude are not recorded linearly in the data files. Signal saturation or "pulse pile-up" effect occurs because of the inability of the counting electronics to discriminate in time a very large number of photon-counts reaching the detector (e.g., Muller, 1973; Donovan et al., 1993). In the present work, we describe the common case of non-paralyzable photon-counting systems, which allows for an analytical correction of the pulse pile-up effect.

Given conditions 1) through 4), the photon counts $P$ reaching the detector of a given channel can be expressed as a function of the discretized raw signal $R$ recorded in the data files at altitude $z(k)$:

$$P(k) = \frac{R(k)}{1 - \tau \frac{c}{2 \delta z L} R(k)} - B(k) \qquad (10)$$

$B$ is the sum of sky and electronic background noise, $\tau$ is the photon-counting hardware dead-time characterizing the pulse pile-up effect (sometimes called resolving time), $c$ the speed of light, and $L$ the number of laser pulses for which the signal was actually recorded in the data files.

5) We then correct the signal for all known altitude-dependent factors according to **Eq. (9)**. For a given channel operating at the emission wavelength $\lambda_1$ and detection wavelength $\lambda_2$ ($\lambda_1$ and $\lambda_2$ are identical for Rayleigh backscatter channels), we then define $N$ as the lidar-measured relative number density, which can be written as a function the saturation-background-corrected signal $P$:

$$N(k) = \frac{(z(k) - z_L)^2}{\eta(k)} P(k) \exp\left( \sum_{k'=0}^{k} \left( (\sigma_{M\_1} + \sigma_{M\_2}) N_a(k') + \sum_{ig} (\sigma_{ig\_1}(k') + \sigma_{ig\_2}(k')) N_{ig}(k') \right) \delta z \right) \qquad (11)$$

In this transformation, the efficiency factor $\eta$ does not have to be known in an absolute manner, but only its variation with altitude range does. Furthermore, if we assume that there is full-overlap between the beam and the telescope field-of-view, then this factor is constant with altitude and does not need to be included at all. The subscript "$M$" and "$ig$" refer to the Rayleigh cross-sections and absorption cross-sections of the interfering gases respectively. The subscripts extensions 1 and 2 refer to the emitted ($\lambda_1$) and received wavelengths ($\lambda_2$) respectively.

With the assumption of full overlap, the lidar-measured relative number density differs from the air number density only by a constant multiplication factor, and therefore does not need to include any of the constant terms with altitude found in the lidar equation as these terms cancel out in the temperature integration process (which implies the ratio of density at two consecutive altitudes).



6) Starting from the top of the profile $z(k_{TOP})$ where temperature is initialized using an ancillary temperature measurement $T_a(k_{TOP})$ (procedure called temperature "tie-on"), the complete temperature profile can be retrieved integrating downward using lidar-measured relative number density. The real-world version of **Eq. (10)** becomes:

$$T(k) = \frac{N(k_{TOP})}{N(k)} T_a(k_{TOP}) + \frac{M_a \delta z}{R_a N(k)} S(k) \tag{12}$$

where $S(k)$ is the discretized version of the summation term in **Eq. (9)**:

$$S(k) = \sum_{k'=k}^{kTOP-1} \overline{N}(k') \overline{g}(k') \tag{13}$$

Like in **Eq. (10)**, the horizontal bar above $N$ and $g$ denotes the mean value of $N$ and $g$ in the vertical layer comprised between $z(k')$ and $z(k'+1)$. The lidar-derived relative density $N$ can be approximated by an exponential function of altitude range, and the layer-averaged density is computed using its geometric mean:

$$\overline{N}(k') = \sqrt{N(k')N(k'+1)} \tag{14}$$

The Earth's gravity field is three-dimensional but its variation with longitude is so small that it can be approximated by a function of latitude and altitude only. For small vertical increments, the variation of $g$ with height is nearly linear, and its layer-averaged value can be expressed as a function of the height $h$ above the reference ellipsoid averaged between $z(k')$ and $z(k'+1)$:

$$\overline{g}(k') = g_0 \left( 1 + g_1 \overline{h}(k') + g_2 \overline{h}^2(k') \right) \tag{15}$$

The height above the reference ellipsoid averaged between $z(k')$ and $z(k'+1)$ takes the form:

$$\overline{h}(k') = \frac{1}{2} \left( h(k') + h(k'+1) \right) \tag{16}$$

The constants $g_0$, $g_1$ and $g_2$ in **Eq. (15)** relate to the Earth's geometry and to the geodetic latitude of the lidar site.

8) As in any real physical measurement, detection noise induces undesired high-frequency noise in the raw lidar
signals. This noise can be reduced by digitally filtering the signals and/or the retrieved temperature profiles. The filtering process impacts the propagation of uncertainties and therefore should be included in the measurement model. When filtering is applied to the lidar signal (i.e., before temperature is computed), the signal's exponential decrease with altitude must be taken into account. For a given altitude $z(k)$, the filtering process in this case therefore consists of convolving a set of filter coefficients $c_p$ with the logarithm of the unsmoothed signal $s_u$ ($s_u=R$ or $s_u=P$ or
$s_u=N$) to obtain a smoothed signal $s_m$ following the expression:

$$s_m(k) = \exp\left( \sum_{p=-n}^{n} c_p(k) \log\left( s_u(k+p) \right) \right) \tag{17}$$

When vertical filtering is applied to the retrieved temperature profile, the filtering process at each individual altitude $z(k)$ consists of convolving the filter coefficients $c_p$ with the unsmoothed temperature $T$ to obtain a smoothed temperature $T_m$: following the expression:

$$T_m(k) = \sum_{p=-n}^{n} c_p(k) T(k+p) \tag{18}$$



In **Eqs. (17)-(18)**, the filter coefficients should be symmetric ($c_p=c_{-p}$ for all $p$) to achieve proper smoothing. Their number and values determine which noise frequencies will be reduced most. A review of digital filtering and recommendations for the use of standardized vertical resolution definitions are provided in our "Part 1" companion paper (**Leblanc et al., 2016a**).

Equations **(10)-(18)** constitute our proposed standardized temperature measurement model. The output quantity is temperature (left-hand side of **Eq. (12)**), while the input quantities are all the variables introduced on the right-hand side of **Eqs. (10)-(15)**. The input quantities' standard uncertainty must be introduced, then propagated through the temperature measurement model, and then combined to produce a temperature combined standard uncertainty profile.

Based on **Eq. (10)**, the instrumentation-related input quantities to consider in the NDACC-lidar standardized temperature uncertainty budget are:

    1) Detection noise inherent to photon-counting signal detection

    2) Saturation (pulse pile-up) correction parameters (typically, photon-counters' dead-time $\tau$)

    3) Background noise extraction parameters (typically, fitting parameters for function $B$)

Based on **Eqs. (11)-(15)**, the additional input quantities to consider in the NDACC-lidar standardized temperature uncertainty budget are:

    4) Rayleigh extinction cross-sections $\sigma_M$

    5) Ancillary air number density profile $N_a$ (or temperature $T_a$ and pressure $p_a$ profiles)

    6) Absorption cross-sections of the interfering gases $\sigma_{ig}$

7) Number density profiles $N_{ig}$ (or mixing ratio profile $q_{ig}$) of the interfering species

    8) Ancillary air temperature for tie-on at the top of the profile $T_a(k_{TOP})$

    9) Acceleration of gravity $g$

    10) The molecular mass of air $M_a$

The interfering gases "$ig$" to consider in practice are ozone and NO$_2$. Because of either very low concentrations or
very low values of their absorption cross-sections, no other atmospheric gases or molecules are known to interfere with the temperature retrieval. The impact of absorption by ozone on the temperature retrieval is very small (<0.1 K) if working at wavelengths near the ozone minimum absorption region (e.g., 355 nm, 387 nm), but can account for up to 1 K error if neglected when working in the Chappuis band (e.g., 532 nm and 607 nm). Conversely, absorption by NO$_2$ is very small for temperature retrievals in the Chappuis band, but can account for up to a 0.2 K error if
neglected at 355 nm and 387 nm.

The contributions of the acceleration of gravity is very small (<0.1 K) providing that the gravity model is altitude-dependent (Lemoine et al., 1998). In the upper mesosphere, the change in the air major species' mixing ratio induces a change with altitude of the air molecular mass and Rayleigh scattering cross-sections. However the induced changes remain below 0.1 K below 90 km, which is much less than the expected uncertainty owed to the other
sources such as detection noise and tie-on temperature uncertainty (Argall, 2007). For temperature profiles reaching 100 km or higher, the change of the molecular mass of air with altitude should be taken into account.



When the receiver field-of-view and the laser beam are known to not fully overlap, an additional "instrumentation-related" uncertainty component must be introduced to take into account the overlap correction (altitude-dependent term $\eta$ in **Eq. (11)**). Because this correction is strongly instrument-dependent, a standardized approach for its treatment cannot be proposed here (beyond the scope of this paper). In the rest of this work, we will therefore

assume full overlap.

The exact altitude of each data bin $k$ can be determined experimentally, for example by tracking the exact position in the data stream of the laser beam backscattering off the laser room hatch (assuming that the receiver and the transmission of the laser beam in the atmosphere are located in the same room). The time (i.e., altitude) resolution of today's lidar data acquisition hardware is very high (of the order of nanoseconds, i.e., a few meters). The exact

altitude of the lidar instrument can also be determined to a precision better than a meter using today's standard geo-positioning methods. For well-designed and well-validated lidar instruments, there is therefore no uncertainty associated with the determination of altitude, and therefore no uncertainty associated with the range correction ($z^2$) term in **Eq. (11)**).

Finally, in our proposed measurement model, uncertainties associated with fundamental physical constants are

neither introduced nor propagated. As described in our "Part 2" companion paper, it is proposed to use fundamental physical constants truncated at a decimal level where no change occurs to its value if adding or subtracting its uncertainty. It is also recommended to use the values reported by the International Council for Science (ICSU) Committee on Data for Science and Technology (CODATA, http://www.codata.org/), endorsed by the BIPM (Mohr et al., 2008). For example, the molar gas constant value $R_a$ reported by the CODATA is 8.3144621 Jmol⁻¹K⁻¹ with an

uncertainty of 0.0000075 Jmol⁻¹K⁻¹. If we truncate to the value of 8.3145 Jmol⁻¹K⁻¹, adding or subtracting its uncertainty does not modify the truncated value, and we therefore consider this value as "exact" (i.e., no uncertainty to be propagated). Note that if the uncertainty of a fundamental constant is of similar order of magnitude as that of some other uncertainty components already identified, then this constant must be included among the input quantities and its uncertainty should be taken into account and propagated just like all other input quantities.

**4 Proposed formulation for the propagation of uncertainty through the lidar temperature retrieval**

In the present section, the law of propagation of uncertainty (**Eq. (2)**) is used to propagate the uncertainty components introduced in our proposed standardized measurement model (previous section). The reader should refer to **section 2** of our "Part 2" companion paper or to the ISSI Team Report (**Leblanc et al., 2016a**) for more details on the conditions of validity of some of the expressions proposed hereafter.

In order to distinguish between the uncertainty source and the quantity for which the uncertainty is calculated, a standardized notation is used throughout this section: Each new equation introduced represents a measurement "sub-model" that yields an output quantity $Y$ with individual uncertainty components $u_{Y(Xi)}$ owed to the uncertainty source $X_i$. Furthermore each introduced component $u_{Y(Xi)}$ is assumed independent from the other components $u_{Y(Xj)}$ ($j{\neq}i$), thus allowing a full description of their covariance matrix in the altitude dimension, and therefore a propagation "in

parallel" with the other independent components throughout signal processing.



### 4.1 Uncertainty owed to detection noise

Signal detection uncertainty is introduced at the detection level, where the signal is recorded in the data files (raw signal $R$). It is derived from a Poisson statistics associated with the probability of detection of a repeated random event (Type-A uncertainty estimation). Using the subscript "($DET$)" for "detection noise", the uncertainty in the raw signal $R$ owed to detection noise expressed for each altitude bin $k$ and for a single temperature channel is written:

$$u_{R(DET)}(k) = \sqrt{R(k)} \qquad (19)$$

There is no correlation between any of the samples considered as this uncertainty component is owed to purely random effects (signal detection). It is propagated to the retrieved temperature profile by systematically assigning the individual input quantities covariance matrix's non-diagonal terms to zero. Assuming a non-paralyzable photon-counting hardware, this uncertainty component is therefore propagated to the saturation and background noise-corrected signal $P$ by applying **Eq. (2)** with no covariance terms to the signal transformation **Eq. (10)**:

$$u_{P(DET)}(k) = \left(\frac{P(k)}{R(k)}\right)^2 \sqrt{R(k)} \qquad (20)$$

This uncertainty component is then propagated to the lidar-derived relative density $N$ by applying **Eq. (2)** to the signal transformation **Eq. (11)**:

$$u_{N(DET)}(k) = \frac{N(k)}{P(k)} u_{P(DET)}(k) \qquad (21)$$

Next, it is propagated through **Eq. (12)** assuming that the signals are uncorrelated between two consecutive altitudes. Applying **Eq. (2)** to the signal transformation **Eq. (14)** yields:

$$u_{\overline{N}(DET)}(k') = \frac{1}{2}\sqrt{\frac{N(k'+1)}{N(k')}u_{N(DET)}^2(k') + \frac{N(k')}{N(k'+1)}u_{N(DET)}^2(k'+1)} \qquad (22)$$

The detection noise uncertainty then propagates to the sum $S$ defined in **Eq. (13)**:

$$u_{S(DET)}(k) = \sqrt{\sum_{k'=k}^{kTOP-1} \overline{g}^2(k') u_{\overline{N}(DET)}^2(k')} \qquad (23)$$

Finally, the temperature uncertainty owed to detection noise $u_{T(DET)}$ is computed by applying **Eq. (2)** to the density integration **Eq. (12)**:

$$u_{T(DET)}(k) = \frac{1}{N(k)T(k)}\sqrt{T^2(k)u_{N(DET)}^2(k) + T_a^2(k_{TOP})u_{N(DET)}^2(k_{TOP}) + \left(\frac{M_a \delta z}{R_a}\right)^2 u_{S(DET)}^2(k)} \qquad (24)$$

The temperature uncertainty owed to detection noise can be of any order of magnitude, depending on altitude and lidar performance and/or specification. **Figure 1** shows this order of magnitude as a function of signal magnitude (left) and altitude (right) for a wide range of lidar specifications. A channel performance is defined here for a given sampling resolution as the altitude at which the signal count rate is 1 MHz. Using such generic representation allows to identify a family of curves, all of which having the same e-folding rate with altitude and signal magnitude. This way, the actual order of magnitude of the temperature uncertainty can be inferred for any lidar system of specific



performance. Not surprisingly, this uncertainty component's e-folding rate is approximately 14 km (black arrow on the right plot), which corresponds to the square-root of the 7-km e-folding rate of air number density. The results are shown for a 120-min integration time and 50 Hz laser repetition rate. For a four-time shorter integration time (30 min), all curves would shift to the right by a factor of 2. For a four-time longer integration time (8 hours), all curves would shift to the left by a factor of 2.

### 4.2 Uncertainty owed to saturation (pulse pile-up) correction

The uncertainty component owed saturation correction depends on the hardware's dead-time $\tau$ and its uncertainty $u_\tau$, which are typically known from the technical specifications provided by the hardware manufacturer (Type-B estimation). This uncertainty component is introduced where the signal is recorded in the data files (raw signal $R$). Using the subscript "($SAT$)" for "saturation", the saturation correction uncertainty propagated to the saturation and background noise-corrected signal $P$ is obtained by applying **Eq. (2)** to the signal transformation **Eq. (10)**:

$$u_{P(SAT)}(k) = \frac{2\delta z}{cL} P^2(k) u_\tau \qquad (25)$$

Just like the detection noise component, the saturation correction uncertainty component is propagated to the lidar-derived relative density $N$ by applying **Eq. (2)** to the signal transformation **Eq. (11)**:

$$u_{N(SAT)}(k) = \frac{N(k)}{P(k)} u_{P(SAT)}(k) \qquad (26)$$

The saturation correction is applied to the lidar signals consistently at all altitudes. Its uncertainty is therefore propagated through **Eq. (12)** assuming full correlation between two consecutive altitudes $z(k')$ and $z(k'+1)$. In these conditions, applying **Eq. (2)** to the signal transformation **Eq. (14)** yields:

$$u_{\overline{N}(SAT)}(k') = \frac{\overline{N}(k')}{2}\left(\frac{u_{N(SAT)}(k')}{N(k')} + \frac{u_{N(SAT)}(k'+1)}{N(k'+1)}\right) \qquad (27)$$

The saturation correction uncertainty then propagates to the sum $S$ defined in **Eq. (13)** assuming again full correlation between altitude bins:

$$u_{S(SAT)}(k) = \sum_{k'=k}^{kTOP-1} g(k') u_{\overline{N}(SAT)}(k') \qquad (28)$$

Finally, the temperature uncertainty owed to saturation correction $u_{T(SAT)}$ is computed by applying **Eq. (2)** to the density integration **Eq. (12)** with the same full correlation assumptions:

$$u_{T(SAT)}(k) = \frac{1}{N(k)}\left| T(k)u_{N(SAT)}(k) - T_a(k_{TOP})u_{N(SAT)}(k_{TOP}) - \frac{M_a\delta z}{R_a} u_{S(SAT)}(k)\right| \qquad (29)$$

**Figure 2** shows the order of magnitude of this uncertainty component as a function of signal strength (left) and altitude (right) for two saturation correction cases, namely if the dead-time is dead-time is 20 ns (max. count rate of 50 MHz, dashed curves), and if the dead-time is 4 ns (max count rate of 250 MHz, solid curves). As for detection noise uncertainty, the results are presented in generic form so that the actual order of magnitude of this uncertainty component can be easily estimated for lidar systems of any performance. Here, the same family of curves is





obtained when the uncertainty is represented as function of the ratio of the signal to the maximum counting rate (left plot).

### 4.3 Uncertainty owed to background noise extraction

Background noise is typically subtracted from the total signal by fitting the uppermost part of the lidar signal with a constant, linear or non-linear function of altitude. An uncertainty component associated with the noise fitting procedure should be introduced. Here we consider the simple case of a linear fit, knowing that the exact same approach can be used for other fitting functions. The linear fitting function to be estimated can be written:

$$B(k) = b_0 + b_1 z(k) \tag{30}$$

For standard fitting methods such as least-squares, the uncertainty $u_{bi}$ and correlation coefficients $r_{bi,bj}$ of the fitting coefficients $b_i$ can be calculated analytically (Type-A estimation) (Press et al., 1986). Using the subscript "($BKG$)" for "background noise", the background noise correction uncertainty can then be introduced by applying **Eq. (2)** to the signal transformation equations **Eq. (10)**:

$$u_{P(BKG)}(k) = \sqrt{u_{b0}^2 + u_{b1}^2 z^2(k) + 2z(k)\,\mathrm{cov}(b_0, b_1)} \tag{31}$$

The above expression can be expanded and/or modified based on the actual form of the fitting function, and taking into account the fitting coefficients' covariance matrix returned by the fitting routine. Just like the saturation correction uncertainty, the uncertainty component owed to the background noise extraction can be propagated through the temperature retrieval assuming full correlation in altitude. Applying **Eq. (2)** to the signal transformations **Eqs. (11)-(14)** therefore yields:

$$u_{N(BKG)}(k) = \frac{N(k)}{P(k)} u_{P(BKG)}(k) \tag{32}$$

$$u_{\overline{N}(BKG)}(k') = \frac{\overline{N}(k')}{2}\left(\frac{u_{N(BKG)}(k')}{N(k')} + \frac{u_{N(BKG)}(k'+1)}{N(k'+1)}\right) \tag{33}$$

$$u_{S(BKG)}(k) = \sum_{k'=k}^{kTOP-1} g(k') u_{\overline{N}(BKG)}(k') \tag{34}$$

$$u_{T(BKG)}(k) = \frac{1}{N(k)}\left| T(k) u_{N(BKG)}(k) - T_a(k_{TOP}) u_{N(BKG)}(k_{TOP}) - \frac{M_a \delta z}{R_a} u_{S(BKG)}(k) \right| \tag{35}$$

The order of magnitude of this uncertainty component depends on the magnitude of the background noise, and if signal-induced noise is present, on the slope of this noise with respect to the signal slope. **Figure 3** shows several examples of constant background noise of varying magnitude. The temperature uncertainty is represented here as a function of altitude (top-left), distance from the tie-on altitude (top-right), signal-to-noise ratio (bottom-left), and statistical uncertainty (bottom-right). The curves show a systematic pattern which consists of a rapid increase in the first 3-4 km below the tie-on altitude as density is integrated downward, followed by a decrease as we get further and further from the tie-on altitude. The e-folding rate is 7 km for the entire family of curves, which reflects the



main influence of the $1/N$ term in **Eq. (35)**. The temperature uncertainty maximum is larger when the magnitude of the noise is larger (as shown for the 387 nm and 607 nm Raman channels on **Fig. 3**).

### 4.4 Uncertainty owed to Rayleigh extinction cross-sections

All lidar-derived relative density uncertainty components owed to the atmospheric extinction are computed by applying **Eq. (2)** to **Eq. (11)**. The Rayleigh extinction cross-sections at the emitted and received wavelengths are among the input quantities. Their values typically originate from theoretical calculations assuming a given atmospheric composition (see for example (Bates, 1984; Eberhard, 2010)), and can be assumed constant with altitude (well-mixed atmosphere). The associated uncertainty, as reported in the literature, is either owed to random or systematic effects, or both. These two types of uncertainty are not introduced and propagated identically in the lidar temperature measurement model. The subscripts suffix "*R*" (for "random") and "*S*" (for "systematic") is used thereafter to make this distinction.

### 4.4.1 Lidar-derived relative density uncertainty for Rayleigh backscatter channels

For Rayleigh backscatter channels, the received wavelength ($\lambda_2$) is identical to the emitted wavelength ($\lambda_1$), and the cross-section uncertainty owed to random and systematic effects is introduced and propagated identically throughout the temperature retrieval. Using the subscript "($\sigma M$)" for "molecular extinction cross-section" uncertainty component, and the suffixes "*R*" and "*S*" for random and "systematic" components respectively, the Rayleigh extinction cross-section uncertainty owed to random and systematic effects can be propagated to the lidar-derived relative density $N$ by applying **Eq. (2)** to **Eq. (11)**:

$$u_{N(\sigma MX)}(k) = 2N(k)\delta z \sum_{k'=0}^{k} N_a(k') u_{\sigma M\_1X} \qquad \text{with } X=R, S \qquad (36)$$

### 4.4.2 Lidar-derived relative density uncertainty for Raman backscatter channels

For Raman backscatter channels (McGee et al., ), the received and emitted wavelengths are different, and the cross-section uncertainty owed to random and systematic effects are introduced and propagated differently. For the uncertainty component owed to random effect, applying **Eq. (2)** to **Eq. (10)** yields:

$$u_{N(\sigma MR)}(k) = N(k)\delta z \sum_{k'=0}^{k} N_a(k') \sqrt{u_{\sigma M\_1R}^2 + u_{\sigma M\_2R}^2} \qquad (37)$$

For the uncertainty component owed to systematic effects, applying **Eq. (2)** to **Eq. (11)** yields:

$$u_{N(\sigma MS)}(k) = N(k)\delta z \sum_{k'=0}^{k} N_a(k') \left( u_{\sigma M\_1S} + u_{\sigma M\_2S} \right) \qquad (38)$$





### 4.4.3 Propagation to temperature

For both Rayleigh and Raman backscatter, both random and systematic components of the lidar-derived relative density uncertainty owed to Rayleigh extinction cross-sections are propagated to temperature similarly to the saturation and background uncertainty components (e.g., **Eqs. (27)-(29)**):

$$u_{\overline{N}(\sigma MX)}(k') = \frac{\overline{N}(k')}{2}\left(\frac{u_{N(\sigma MX)}(k')}{N(k')} + \frac{u_{N(\sigma MX)}(k'+1)}{N(k'+1)}\right) \qquad \text{with } X=R, S \tag{39}$$

$$u_{S(\sigma MX)}(k) = \sum_{k'=k}^{kTOP-1} g(k') u_{\overline{N}(\sigma MX)}(k') \qquad \text{with } X=R, S$$

$$u_{T(\sigma MX)}(k) = \frac{1}{N(k)}\left| T(k)u_{N(\sigma MX)}(k) - T_a(k_{TOP})u_{N(\sigma MX)}(k_{TOP}) - \frac{M_a \delta z}{R_a} u_{S(\sigma MX)}(k) \right| \qquad \text{with } X=R, S \tag{40}$$

The magnitude of the uncertainty owed to the Rayleigh cross-sections is plotted in **Fig. 4** for four different wavelengths and for components owed to both systematic and random effects. The results are shown for each 1% cross-section uncertainty, i.e., if the cross-section is introduced in the lidar measurement model with a 5% uncertainty, then the temperature uncertainty will be five times larger than shown in **Fig. 4**. Again for all curves, the e-folding rate is 7 km, which reflects the dominant influence of the term $1/N$ is **Eq. (40)**.

### 4.5 Uncertainty owed to air number density

An ancillary profile of air number density ($N_a$) is needed to correct for Rayleigh extinction as formulated in **Eq. (11)**. Air number density is generally not estimated directly, but rather derived from air temperature and pressure. First we will consider the case of number density being the input quantity, then we will consider the case of temperature and pressure being the input quantities.

### 4.5.1 If air number density is the input quantity

Here, it is assumed that the air density profile $N_a$ is made of fully-correlated values in altitude. If air number density is not derived from air temperature and pressure, its uncertainty $u_{Na}$ is propagated to the lidar-derived relative density by applying **Eq. (2)** to **Eq. (11)** in a straightforward manner:

$$u_{N(Na)}(k) = N(k)\delta z\left(\sigma_{M\_1} + \sigma_{M\_2}\right)\sum_{k'=0}^{k} u_{Na}(k') \tag{41}$$

This component is then propagated to temperature using the same approach as for saturation and background noise correction uncertainties.

### 4.5.2 If air temperature and pressure are the input quantities

When the ancillary number density is computed from an ancillary temperature $T_a$ and pressure $p_a$ source (e.g., radiosonde measurements or meteorological models), the uncertainties $u_{Ta}$ and $u_{pa}$ must be introduced and the degree of correlation between temperature and pressure must be estimated.



If temperature and pressure are measured or computed independently, then the complete covariance matrix in the vertical dimension needs to be estimated. This is the most complex case to consider because of the interplay between the lack of correlation between $T_a$ and $p_a$ at any given altitude, and the high correlation between the temperature values at two consecutive altitudes, or between the pressure values at two consecutive altitudes. However, a good approximation consists of considering the propagation linear, i.e., first combining the uncertainties at one fixed level assuming no correlation, and then propagating the combined uncertainty assuming full correlation between two consecutive altitudes. In this case, the lidar-derived relative density uncertainty owed to the ancillary air number density can be written:

$$u_{N(Na)}(k) = N(k)\delta z \sum_{k'=0}^{k} \left( \sigma_{M\_1} + \sigma_{M\_2} \right) N_a(k') \sqrt{\frac{u_{pa}^2(k')}{p_a^2(k')} + \frac{u_{Ta}^2(k')}{T_a^2(k')}} \tag{42}$$

If temperature and pressure are known to be fully correlated, then, the lidar-derived relative density uncertainty owed to the ancillary air number density becomes:

$$u_{N(Na)}(k) = N(k)\delta z \sum_{k'=0}^{k} \left( \sigma_{M\_1} + \sigma_{M\_2} \right) N_a(k') \left| \frac{u_{pa}(k')}{p_a(k')} - \frac{u_{Ta}(k')}{T_a(k')} \right| \tag{43}$$

### 4.5.3 Propagation to temperature

The lidar-derived number density uncertainty owed to ancillary air number density is propagated to temperature by applying **Eq. (2)** to **Eqs. (12)-(14)** assuming full correlation in altitude:

$$u_{\overline{N}(Na)}(k') = \frac{\overline{N}(k')}{2} \left( \frac{u_{N(Na)}(k')}{N(k')} + \frac{u_{N(Na)}(k'+1)}{N(k'+1)} \right) \tag{44}$$

$$u_{S(Na)}(k) = \sum_{k'=k}^{kTOP-1} g(k') u_{\overline{N}(Na)}(k') \tag{45}$$

$$u_{T(Na)}(k) = \frac{1}{N(k)} \left| T(k) u_{N(Na)}(k) - T_a(k_{TOP}) u_{N(Na)}(k_{TOP}) - \frac{M_a \delta z}{R_a} u_{S(Na)}(k) \right| \tag{46}$$

**Figure 5** shows the magnitude of this uncertainty component assuming either that the input quantity is air number density (left plot), or that the input quantities are temperature and pressure (right plot). In the first case, the results are shown for 1% uncertainty in ancillary air number density. In the second case, the results are plotted for 1 K ancillary temperature uncertainty (solid curves) and 0.1 hPa ancillary pressure uncertainty (dash curves). The shape of the dash curves do not show the normal 7-km e-folding rate because of the emerging very high pressure relative uncertainty associated with a fixed 0.1 hPa value. The e-folding rate would be similar to the other curves if the ancillary pressure uncertainty was set to be constant in relative value rather than absolute.

### 4.6 Uncertainty owed to the ozone and NO$_2$ absorption cross-sections

Temperature-dependent ozone and NO$_2$ absorption cross-section values typically can be found in published works originating from spectroscopy groups around the world (e.g., Brion et al., 1998; Bogumil, 2003; Chehade et al.,





2013, 2003; Gorshelev et al., 2014; Burkholder and Talukdar, 1994; Burrows et al., 1999; Vandaele et al., 1998). The random component of the cross-section uncertainty is normally provided in these works. Occasionally, one or more components owed to systematic effects are also provided. Just like for Rayleigh extinction cross-sections, these two types of component are not introduced and propagated identically in the lidar temperature measurement model. The formulation of their propagation is identical to that just presented for Rayleigh extinction cross-sections (**Eqs. (36)-(40)**), except that the air number density is replaced by the interfering gas number density, and the cross-section uncertainty is now a function of temperature, i.e., altitude.

For Rayleigh backscatter channels:

$$u_{N(\sigma igX)}(k) = 2N(k)\delta z \sum_{k'=0}^{k} N_{O3}(k') u_{\sigma ig\_1X}(k') \qquad \text{with } ig = O_3, NO_2 \text{ and } X=R,S \qquad (47)$$

For Raman backscatter channels:

$$u_{N(\sigma igR)}(k) = N(k)\delta z \sqrt{\sum_{k'=0}^{k} N_{ig}^2(k')\left(u_{\sigma ig\_1R}^2(k') + u_{\sigma ig\_2R}^2(k')\right)} \qquad \text{with } ig = O_3, NO_2 \qquad (48)$$

$$u_{N(\sigma igS)}(k) = N(k)\delta z \sum_{k'=0}^{k} N_a(k')\left(u_{\sigma ig\_1S}(k') + u_{\sigma ig\_2S}(k')\right) \qquad \text{with } ig = O_3, NO_2 \qquad (49)$$

Their propagation to temperature can then be written:

$$u_{\overline{N}(\sigma igX)}(k') = \frac{\overline{N}(k')}{2}\left(\frac{u_{N(\sigma igX)}(k')}{N(k')} + \frac{u_{N(\sigma igX)}(k'+1)}{N(k'+1)}\right) \qquad \text{with } ig = O_3, NO_2 \text{ and } X=R,S \qquad (50)$$

$$u_{S(\sigma igX)}(k) = \sum_{k'=k}^{kTOP-1} g(k') u_{\overline{N}(\sigma igX)}(k') \qquad \text{with } ig = O_3, NO_2 \text{ and } X=R,S \qquad (51)$$

$$u_{T(\sigma igX)}(k) = \frac{1}{N(k)}\left| T(k) u_{N(\sigma igX)}(k) - T_a(k_{TOP}) u_{N(\sigma igX)}(k_{TOP}) - \frac{M_a \delta z}{R_a} u_{S(\sigma igX)}(k)\right| \quad ig = O_3, NO_2; X=R,S \quad (52)$$

The magnitude of this uncertainty component owed to both systematic and random effects is shown in **Fig. 6** for both Rayleigh and Raman backscatter cases and different wavelengths. The contribution of ozone absorption (left plot) is larger in the visible (532 nm and 607 nm which are both in the Chappuis band) than in the ultraviolet (355 nm and 387 nm). Conversely, the contribution of $NO_2$ absorption (right plot) is larger in the ultraviolet than in the visible.

**4.7 Uncertainty owed to ancillary ozone and NO$_2$ number density profiles**

The ozone and $NO_2$ absorption terms in **Eq. (11)** comprise the sum of ancillary ozone and $NO_2$ number densities taken at all altitudes from the ground to the altitude considered $z(k)$. Depending on the data source, these ancillary profiles may be mixing ratio or number density (Ahmad et al., 1987; Bauer et al., 2012; Bracher et al., 2005; Brohede et al., 2007) Assuming that all values within the same ancillary profile are fully correlated, uncertainty components owed to the ancillary ozone and $NO_2$ profiles can be propagated to temperature similarly to the uncertainty component owed to air number density (i.e., **Eq. (41)** and **Eqs. (44)-(46)**). :



$$u_{N(Nig)}(k) = N(k)\sum_{k'=0}^{k}\left(\sigma_{ig\_1}(k') + \sigma_{ig\_2}(k')\right)u_{Nig}(k') \qquad \text{with } ig = O_3, NO_2 \qquad (53)$$

$$u_{\overline{N}(Nig)}(k') = \frac{\overline{N}(k')}{2}\left(\frac{u_{N(Nig)}(k')}{N(k')} + \frac{u_{N(Nig)}(k'+1)}{N(k'+1)}\right) \qquad (54)$$

$$u_{S(Nig)}(k) = \sum_{k'=k}^{kTOP-1} g(k') u_{\overline{N}(Nig)}(k') \qquad (55)$$

$$u_{T(Nig)}(k) = \frac{1}{N(k)}\left| T(k)u_{N(Nig)}(k) - T_a(k_{TOP})u_{N(Nig)}(k_{TOP}) - \frac{M_a\delta z}{R_a}u_{S(Nig)}(k)\right| \qquad (56)$$

5 **Figure 7** shows the magnitude of this uncertainty component for both ozone (left) and $NO_2$ (right), for both Rayleigh and Raman backscatter cases, and for different wavelength bands. The results are shown per 1% ancillary ozone and $NO_2$ uncertainty (solid curves), and per 1 ppmv ancillary ozone (respectively 1 ppbv ancillary $NO_2$) uncertainty (dash curves). Similarly to the temperature uncertainty owed to the absorption cross-sections, the contribution of ozone is larger in the visible than in the UV, and the contribution of $NO_2$ is larger in the UV than in 10 the visible.

### 4.8 Uncertainty owed to the temperature tie-on at the top of the profile

Equation **(12)** shows that an ancillary temperature value $T_a$ at altitude $z(k_{TOP})$ is needed to initialize the profile at the top. Using the subscript "(*TIE*)" for "tie-on", the ancillary temperature uncertainty $u_{Ta}(k_{TOP})$ is propagated to the retrieved temperature profile by applying **Eq. (2)** to **Eq. (12)**:

15 $$u_{T(TIE)}(k) = \frac{N(k_{TOP})}{N(k)}u_{Ta}(k_{TOP}) \qquad (57)$$

The magnitude of this uncertainty component is plotted in **Fig. 8** for a 1 K tie-on ancillary temperature uncertainty and for several lidar performance cases. As expected, we obtain a family of curves with an approximate e-folding rate of 7 km due to the term $1/N$ in **Eq. (57)**.

### 4.9 Uncertainty owed to the acceleration of gravity

20 The acceleration of gravity is an input quantity introduced in **Eq. (13)**. The constants $g_0$, $g_1$ and $g_2$ relate to the Earth's geometry and to the geodetic latitude of the lidar site. If a value of the local ellipsoid height at the lidar site $h(0)$ is not known, we can approximate it to the site's altitude above mean sea level $z(0)$. For all altitude-dependent and latitude-dependent formulations of the acceleration of gravity, the difference between $h(0)$ and $z(0)$ is by far the largest source of error in the computation of the acceleration of gravity. We therefore can define a new uncertainty 25 component $u_h$ associated with the approximation of $h$. The values of $h$ at neighboring altitudes are fully correlated, and their standard uncertainty can be deduced directly from **Eq. (16)**:

$$u_{\overline{h}}(k') = \frac{1}{2}\left(u_h(k') + u_h(k'+1)\right) \qquad (58)$$

The height uncertainty is then propagated to temperature by applying **Eq. (2)** to **Eqs. (12)-(15)**:



$$u_{T(g)}(k) = \frac{1}{N(k)} \frac{M_a \delta z}{R_a} g_0 \sum_{k'=k}^{kTOP-1} \overline{N}(k') \left(g_1 + 2g_2 \overline{h}(k')\right) u_{\overline{h}}(k') \qquad (59)$$

**Figure 9** shows the magnitude of this uncertainty component per 0.1% uncertainty in the acceleration of gravity. The results are shown in Kelvin and as a function of altitude (right plot), but also in percent and as a function of the distance from the tie-on altitude (left plot) to illustrate the direct relationship between gravity relative uncertainty and temperature relative uncertainty.

### 4.10 Uncertainty owed to the molecular mass of air

The molecular mass of dry air $M_a$ is introduced in **Eq. (12)**. Its uncertainty $u_{Ma}$, can be propagated to temperature using:

$$u_{T(Ma)}(k) = \frac{\delta z}{R_a} \frac{S(k)}{N(k)} u_{Ma} \qquad (60)$$

This component remains negligible below 90 km, and has a variation with altitude exactly similar to that shown for the acceleration of gravity. Error! Reference source not found. **9** can therefore be used for the molecular mass of air without any change to it.

### 4.11 Propagation of uncertainty when merging multiple channels together

Temperature lidar instruments are usually designed with multiple channels of varying signal intensity to maximize the overall altitude range of the profile. Signal intensity can be reduced using neutral density filters or optical systems attenuating the Rayleigh-backscattered signals, or using Raman backscatter, which intensity is typically 750 times weaker than that of Rayleigh backscatter. Here, the propagation of uncertainty is further considered for at least two channels being merged to form a single profile. Merging individual intensity channels into a single profile can be done either during lidar signal processing or after the temperature is calculated for each individual channel.

### 4.11.1 Merging the temperature profiles retrieved for individual channels

A single profile covering the entire useful range of the instrument is typically obtained by combining the most accurate overlapping sections of the profiles retrieved from individual channels. The thickness of the transition region can vary from a few meters to a few kilometers, depending on the instrument and on the intensity of the channels considered. Considering a low-intensity channel $i_L$ and a high-intensity channel $i_H$, and assuming that the transition region's bottom and top altitudes are $z(k_1)$ and $z(k_2)$ respectively, the merged temperature $T_M$ at any altitude bin $k$ comprised between $k_1$ and $k_2$ is typically obtained by computing a weighted average of the temperature values retrieved for each range $T_L$ and $T_H$ at the same altitude bin:

$$T_M(k) = w(k)T_L(k) + (1 - w(k))T_H(k) \qquad \qquad k_1 \leq k \leq k_2 \text{ and } 0 \leq w(k) \leq 1 \quad (61)$$

The uncertainty components owed to detection noise is propagated assuming no correlation between the two channels:

$$u_{TM(DET)}(k) = \sqrt{\left(w(k)u_{TL(DET)}(k)\right)^2 + \left((1 - w(k))u_{TH(DET)}(k)\right)^2} \qquad k_1 \leq k \leq k_2 \text{ and } 0 \leq w(k) \leq 1 \quad (62)$$





For all uncertainty components that are not of instrumental origin, full correlation is assumed between the two channels:

$$u_{TM(X)}(k) = w(k)u_{TL(X)}(k) + (1 - w(k))u_{TH(X)}(k) \qquad k_1 \leq k \leq k_2 \text{ and } 0 \leq w(k) \leq 1 \quad (63)$$

with $X = \sigma MR, \sigma MS, Na, \sigma igR, \sigma igS, Nig, g, TTOP, Ma$.

For the uncertainty components of instrumental origin (namely, the saturation correction and background noise extraction) the degree of correlation between the channels hardware needs to be estimated. If the two channels use different hardware, they can be assumed independent and the  temperature uncertainties owed to saturation correction and background noise extraction can be written:

$$u_{TM(X)}(k) = \sqrt{(w(k)u_{TL(X)}(k))^2 + ((1 - w(k))u_{TH(X)}(k))^2} \quad k_1 \leq k \leq k_2 \text{ and } 0 \leq w(k) \leq 1 \qquad (64)$$

with $X = SAT, BKG$

If the two channels share the same hardware and if the saturation and background noise corrections have been applied consistently for both channels within the same data processing algorithm, the associated uncertainty components can be propagated to the combined profile assuming full correlation:

$$u_{TM(X)}(k) = w(k)u_{TL(X)}(k) + (1 - w(k))u_{TH(X)}(k) \qquad k_1 \leq k \leq k_2 \text{ and } 0 \leq w(k) \leq 1 \quad (65)$$

with $X = SAT, BKG$.

### 4.11.2 Merging lidar signals before the temperature  profile is computed

The merging procedure can be done on the raw signals ($s=R$), the saturation-background corrected signals ($s=P$), or the lidar-derived relative density ($s=N$). The signals of the channels to combine are of different magnitude. The signal normalization of one channel with respect to the other is therefore necessary before combining the channels.

In addition, the signals decrease with altitude is nearly exponential. The merging procedure should therefore be done on the logarithm of the signal rather than the signal itself. Under these conditions, the merged signal $s_M$ obtained from a low intensity channel signal $s_L$ and a high-intensity channel signal $s_H$ with a normalization constant $\kappa$ is written:

$$s_M(k) = \exp(w(k)\log(s_L(k)) + (1 - w(k))\log(\kappa s_H(k))) \qquad k_1 \leq k \leq k_2 \text{ and } 0 \leq w(k) \leq 1 \quad (66)$$

Similarly to the case of merging the temperature profiles, the uncertainty components owed to detection noise is propagated assuming no correlation between the two channels:

$$u_{sM(SDET)}(k) = s_M(k)\sqrt{\left(w(k)\frac{u_{sL(DET)}(k)}{s_L(k)}\right)^2 + \left((1 - w(k))\frac{u_{sH(DET)}(k)}{s_H(k)}\right)^2} \quad k_1 \leq k \leq k_2 \text{ and } 0 \leq w(k) \leq 1 \quad (67)$$

If the signal to be merged is the lidar-derived relative density ($s=N$), all uncertainty components owed to atmospheric extinction propagate to the merge density using:

$$u_{sM(X)}(k) = s_M(k)\left(w(k)\frac{u_{sL(X)}(k)}{s_L(k)} + (1 - w(k))\frac{u_{sH(X)}(k)}{s_H(k)}\right) \qquad k_1 \leq k \leq k_2 \text{ and } 0 \leq w(k) \leq 1 \quad (68)$$

with $X = \sigma MR, \sigma MS, Na, \sigma igR, \sigma igS, Nig$.



Just like for the case of merging temperature, the degree of correlation between the channels hardware needs to be estimated before we can use a specific formulation for the propagation of the uncertainty components of instrumental origin. If the two channels use different hardware, they can be assumed independent and the merged signal uncertainties owed to saturation correction and background noise extraction can be written

$$u_{sM(SX)}(k) = s_M(k)\sqrt{\left(w(k)\frac{u_{sL(X)}(k)}{s_L(k)}\right)^2 + \left((1-w(k))\frac{u_{sH(X)}(k)}{s_H(k)}\right)^2} \qquad (69)$$

with $X = SAT, BKG$.

If the two channels share the same hardware and if the saturation and background noise corrections have been applied consistently for both channels within the same data processing algorithm, the associated uncertainty components can be propagated to the combined profile assuming full correlation:

$$u_{sM(X)}(k) = s_M(k)\left(w(k)\frac{u_{sL(X)}(k)}{s_L(k)} + (1-w(k))\frac{u_{sH(X)}(k)}{s_H(k)}\right) \qquad (70)$$

with $X = SAT, BKG$.

The uncertainty components owed to temperature tie-on, acceleration of gravity, and the molecular mass of air are not included in the above expressions because they are introduced later in the data processing. In this case, **Eqs. (57)-(60)** apply directly to the temperature profile retrieved from the merged lidar-derived number density.

### 4.12 Temperature combined standard uncertainty

Now that all the independent uncertainty components considered in our lidar temperature measurement model have been reviewed and propagated, we can combine them into a unique temperature combined standard uncertainty:

$$u_T(k) = \sqrt{\begin{array}{l} u_{T(DET)}^2(k) + u_{T(SAT)}^2(k) + u_{T(BKG)}^2(k) + u_{T(TTOP)}^2(k) \\ + u_{T(\sigma MR)}^2(k) + u_{T(\sigma MRS)}^2(k) + u_{T(Na)}^2(k) + u_{T(g)}^2(k) + u_{T(Ma)}^2(k) \\ + u_{T(\sigma O3R)}^2(k) + u_{T(\sigma O3S)}^2(k) + u_{T(NO3)}^2(k) + u_{T(\sigma NO2R)}^2(k) + u_{T(\sigma NO2S)}^2(k) + u_{T(NNO2)}^2(k) \end{array}} \qquad (71)$$

At the tie-on altitude $z(k_{TOP})$, all uncertainty components should be set to zero except uncertainty owed to the ancillary temperature $u_{T(TTOP)}$. Also, when using multiple channels, the temperature combined standard uncertainty should not be computed for individual intensity channels and then merged into a single profile. Instead, the individual uncertainty components should first be propagated to the merged temperature profile and then added in quadrature to obtain the combined standard uncertainty.

If combining multiple profiles measured by the same instrument, for example to compute a climatology, uncertainty components owed to systematic effects in altitude and/or time must remain separated from components owed to random effects. Uncertainty owed to detection noise is always added in quadrature, but for other components, knowledge of the covariance matrix in the time and/or altitude dimension(s) is required (type-A or type-B estimation). It is therefore strongly recommended to always keep each individual component, and **Eq. (71)** should be used only as a "final product".





### 5 Example of actual temperature uncertainty budget

The uncertainty components discussed in the previous section were quantitatively reviewed, for most cases, in parametric form, so that the order of magnitude of each component could be estimated for a wide range of instrument performance. Here we provide an actual example using existing measurements from the Jet Propulsion

Laboratory (JPL) stratospheric ozone DIAL at the NDACC site of Mauna Loa Observatory, Hawaii (MLO). In this example, the input quantities' uncertainty estimates are taken from the JPL in-house data processing software used to process the routine JPL lidar data archived at NDACC. A list of those input quantities and their uncertainty is compiled in **Table**.

The full temperature uncertainty budget is shown in **Fig. 10** for a 2-hour measurement obtained on March 13, 2009.

The results are presented for a typical variable vertical filtering scheme that accommodates the signal magnitude of the different channels yielding a vertical resolution comprised between 0.3 km (lower stratosphere) and 5 km (upper mesosphere). The JPL lidar at MLO comprises 3 ranges (Rayleigh high-intensity, Rayleigh low-intensity, and Raman), and the figures show the uncertainty profiles for each of them (top-left, top-right and bottom-left) as well as for the merged profile (bottom-right). The altitudes of transition from one range to another can be identified by

looking at the magnitude of the uncertainty owed to saturation correction or to detection noise which are signal-dependent (light green and red curves respectively). The transition between the Raman channel and the Rayleigh low-intensity channel is at 31 km, and the transition between the Rayleigh low-intensity channel and Rayleigh high-intensity channel is at 33 km.

The combined standard uncertainty of the merged temperature profile (bottom-right plot, black dash curve) is

obtained by first computing the merged profiles of the individual uncertainty components, and then by combining the merged individual components into a single merged total uncertainty profile. The combined uncertainty curves of the individual channels (dash black curves in the top-left, top-right and bottom left plots) should not be used to compute a combined standard uncertainty for the merged profile.

After optimal combination of all three channels, the temperature standard uncertainty results mainly from 4

components. At the very bottom (10-15 km) the dominant source is the Rayleigh cross-section (0.6 K at 10 km, dark blue curve), then becomes the detection noise of the Raman channel (up to 1 K at 30 km) and the low intensity Rayleigh channel (0.9 K at 31-33 km). After transitioning to the Rayleigh high-intensity channel, uncertainty is equally shared (0.7 K at 33 km) by saturation correction (green) and detection noise (red). Detection noise is then the dominant source of uncertainty all the way up to 75 km (~1.5 K), where it slowly gives way to the tie-on

temperature uncertainty (grey curve), which increases to 20 K at the tie-on altitude.

When using the long-term database accumulated at MLO over the course of 20 years (for example to compute a climatology or to infer interannual variability or trends), uncertainty should be propagated to the climatology by first propagating each individual uncertainty component (the colored curves in the bottom-right plot) and then combining them. The detection noise (red curve) will be added in quadrature while all other components are expected to be

propagated linearly.





## 6 Conclusion

As part of three companion papers that reviewed the recommendations made to the NDACC lidar community for the standardization of vertical resolution and uncertainty, the present article covered the temperature uncertainty budget. The parameters impacting the lidar temperature retrieval include a number of atmospheric species, their scattering or

absorption properties, as well as instrumental specifications. There is therefore no unique expression of uncertainty in the temperature lidar data processing algorithms that can be recommended. However, efforts were made here to produce generic recommendations that can be followed within the entire network.

The recommended definition of uncertainty is combined standard uncertainty as defined by the BIPM (JCGM 200: 2012; JCGM 100: 2008). One important aspect of our proposed approach is the ability to propagate all independent

uncertainty components in parallel through the data processing chain. The individual uncertainty components are then combined together to form the combined standard temperature uncertainty, a mandatory variable of the proposed standardized NDACC lidar temperature uncertainty budget.

The individual uncertainty components identified herein comprise signal detection uncertainty, uncertainty due to saturation correction, background noise extraction, the merging of multiple channels, the absorption cross-sections

of ozone and $NO_2$, the molecular extinction cross-sections, the a priori use of ancillary air, ozone, and $NO_2$ number density, the a priori use of ancillary temperature to tie-on the top of the profile, the acceleration of gravity, and the molecular mass of air. All these sources of uncertainty except detection noise imply correlated terms in the vertical dimension, which means that covariance terms must be taken into account when vertical filtering is applied.

The expression of the individual uncertainty components and their step-by-step propagation through the temperature

data processing chain were thoroughly estimated by the ISSI Team and reviewed here. The proposed formulations were quantitatively verified using simulated lidar signals and Monte Carlo experiments. This validation exercise, which details are provided in the ISSI Team Report (**Leblanc et al., 2016a**), allowed the quantification of each uncertainty component propagated to the retrieved temperature profile in the presence of correlated variables.

It is strongly recommended that every source of uncertainty be reported in the NDACC-archived metadata files. In

addition, individual standard uncertainty components contributing to the temperature combined uncertainty should be reported in the NDACC-archived data files if at all possible. For each reported uncertainty source, the systematic or random components should be reported in both the altitude and time dimensions. If using multiple profiles originating from the same instrument (for example to compute a climatology), the temperature uncertainty should be propagated to the end product by first propagating each individual uncertainty component, and then by combining

them. In this process, the temperature uncertainty owed to detection noise will be added in quadrature while all other uncertainty components are expected to be propagated linearly.

Due to the large variety of instrumentation, some uncertainty component may have not been treated in the present article. For those sources not treated here, the same generic approach as that proposed here should be used, and the individual components should be included to the uncertainty budget presented here following the same propagation

principles.

As mentioned in our "Part 1" companion paper (**Leblanc et al., 2016b**), many concepts described for ozone and temperature in our three companion papers can be applied to the retrieval of other NDACC lidar species such as



water vapor (Raman and differential absorption techniques), temperature (rotational Raman technique, (Arshinov et al., 1983)), and aerosol backscatter ratio. The ISSI Team recommends the formation of new working groups for that purpose.

**Acknowledgements**

5    This work was initiated in response to the 2010 Call for International Teams of Experts in Earth and Space Science by the International Space Science Institute (ISSI) in Bern, Switzerland. It could not have been performed without the travel and logistical support of ISSI. Part of the work described in this paper was carried out at the Jet Propulsion Laboratory, California Institute of Technology, under agreements with the National Aeronautics and Space Administration. Part of this work was carried out in support of the VALID Project. RJS would like to acknowledge

10   the support of the Canadian National Sciences and Engineering Research Council for support of The University of Western Ontario lidar work.



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



**Table 1: List of most commonly used backscatter temperature lidar wavelengths**

| $\lambda_l$ (nm) | $\lambda_2$ (nm) | Backscatter technique | Domain of validity | Light source details ($\lambda_l$) |
|---|---|---|---|---|
| 353 | 353 | Rayleigh | 30 < z < 100 km | Excimer XeCl 308 nm Raman-shifted |
| 353 | 385 | $N_2$ Raman | 10 < z < 40 km | Excimer XeCl 308 nm Raman-shifted |
| 355 | 355 | Rayleigh | 30 < z < 100 km | Nd:YAG tripled 355 nm non-shifted |
| 355 | 387 | $N_2$ Raman | 10 < z < 40 km | Nd:YAG tripled 355 nm non-shifted |
| 532 | 532 | Rayleigh | 30 < z < 110 km | Nd:YAG doubled 532 nm non-shifted |
| 532 | 608 | $N_2$ Raman | 10 < z < 40 km | Nd:YAG doubled 532 nm non-shifted |

**Table 2: Input quantities, and their uncertainty, used to compute the temperature uncertainty budget presented in Fig. 10**

| Input quantity | Dataset Name | Domain of validity | Uncertainty estimate (random) | Reference | Uncert. name | Uncert. used here |
|---|---|---|---|---|---|---|
| $\sigma_M$ | Eberhard | / | 2% | Eberhard, 2010 | $u_{\sigma M}$ | 2% |
| $T_a$ | MSISE-90 NCEP-NDSC Radiosonde | > 47 km 30-47 km < 30 km | 20 K 1-5 K 0.2-0.5 K | Hedin, 1991 Finger et al., 1993 Hurst et al., 2011 | $u_{Ta}$ | 20 K 5 K 0.5 K |
| $p_a$ | MSISE-90 NCEP-NDSC Radiosonde | > 47 km 30-50 km < 30 km | 5% 5% 0.3 hPa | Hedin, 1991 Finger et al., 1993 Hurst et al., 2011 | $u_{pa}$ | 5% 5% 0.1 hPa |
| $\sigma_{O3}$ | DMB | 350-830 nm | 5% | Brion et al., 1998 | $u_{\sigma O3}$ | 5% |
| $N_{O3}$ | WACCM | 30-100 km | 10% | Garcia et al., 2007 | $U_{NO3}$ | 10% |
| $\sigma_{NO2}$ | Bogumil | 200-800 nm | 3.5% | Bogumil et al., 2003 | $u_{\sigma NO2}$ | 5% |
| $N_{NO2}$ | WACCM | 10-120 km | 10% | Garcia et al., 2007 | $U_{NNO2}$ | 10% |
| $T_a(k_{TOP})$ | MSISE-90 NCEP-NDSC | > 47 km < 47 km | 20 K 5 K | Hedin, 1991 Finger et al., 1993 | $u_{TTOP}$ | 20 K 5 K |
| $g$ | WGS-84 | 10-120 km | 0.002% | NIMA, 2000 | $u_g$ | 0.002% |
| $M_a$ | CPIM-2007 | 10-120 km | 0.02% | CPIM-2007 | $u_{Ma}$ | 0.02% |



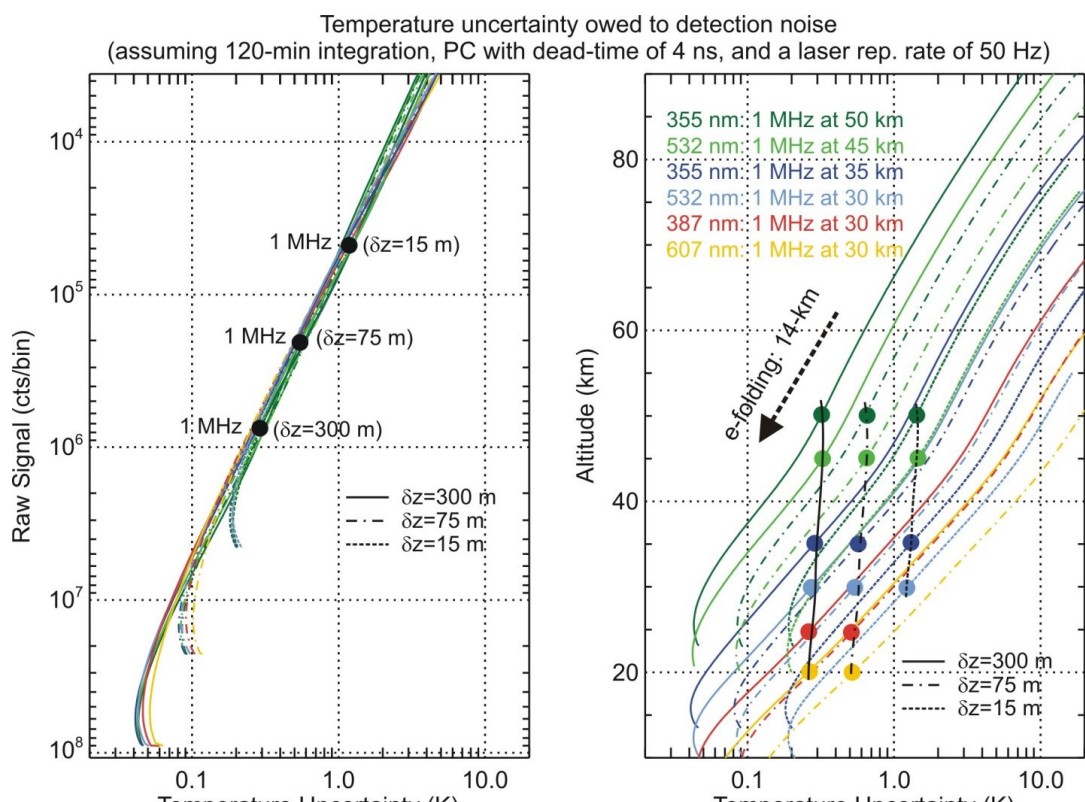

**Figure 1: Temperature uncertainty owed to detection noise as a function of lidar signal magnitude (left) and altitude (right) for a variety of lidar performance configurations, specifically: 2 different signal strengths (1 MHz in the upper stratosphere and 1 MHz in the lower stratosphere), 2 different emission wavelengths (UV and green), 3 different vertical samplings (15 m, 75 m, and 300 m), and two types of backscatter (Rayleigh and Raman). The solid circles indicate the location of the 1 MHz signal count rate for a specific channel.**





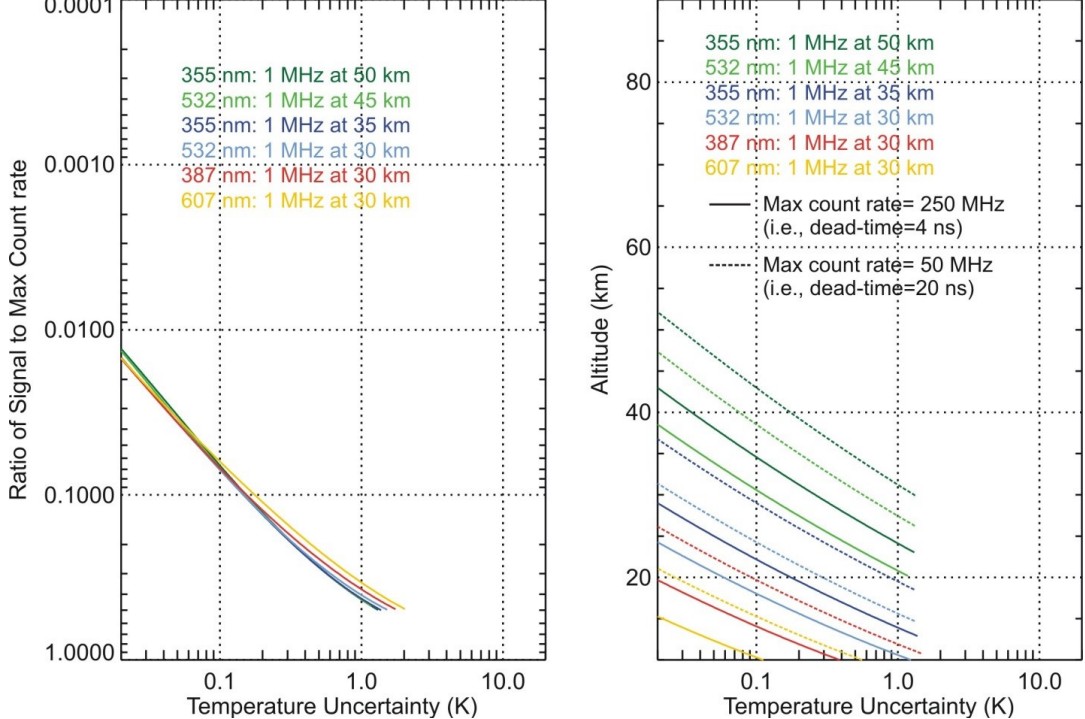

**Figure 2: Temperature uncertainty owed to saturation correction as a function of lidar signal magnitude (left) and altitude (right) for a variety of lidar performance configurations (see Figure 1 caption for details)**





**Figure 3: Temperature uncertainty owed to background noise correction as a function of altitude (top-left), distance from tie-on (top-right), signal-to-noise ratio (bottom left), and statistical uncertainty (bottom-right), for a variety of signal and noise strengths (see Figure 1 caption for details).**





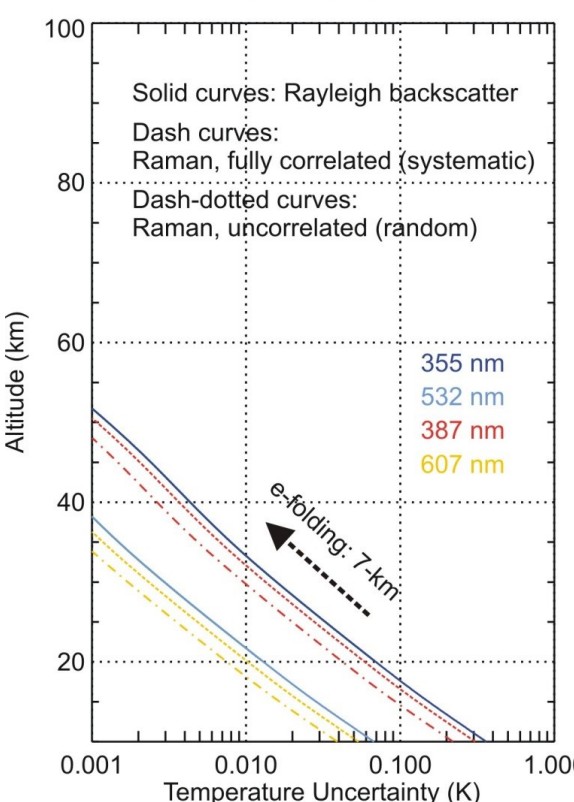

**Figure 4: Temperature uncertainty owed to the Rayleigh cross-section used in the molecular extinction correction. The results are shown per 1% in cross-section uncertainty.**







Figure 5: **Temperature uncertainty owed to the a priori use of ancillary air number density in the molecular extinction correction. The results are shown per 1% ancillary number density uncertainty (left plot), and per 1 K and 0.1 hPa ancillary temperature and pressure uncertainty respectively (right plot).**



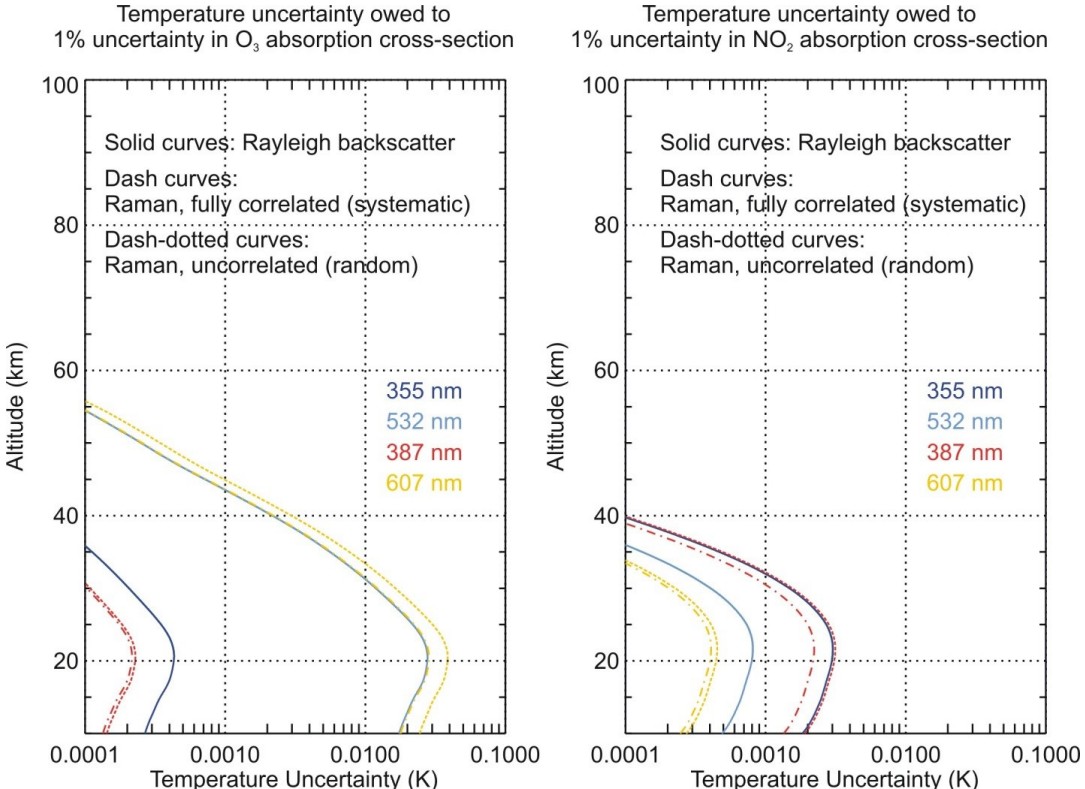

**Figure 6: Temperature uncertainty owed to the cross-sections used for the ozone and NO$_2$ absorption correction. The results are shown per 1% in cross-section uncertainty (left side: ozone, right side: NO$_2$), and for components owed to both systematic and random effects**





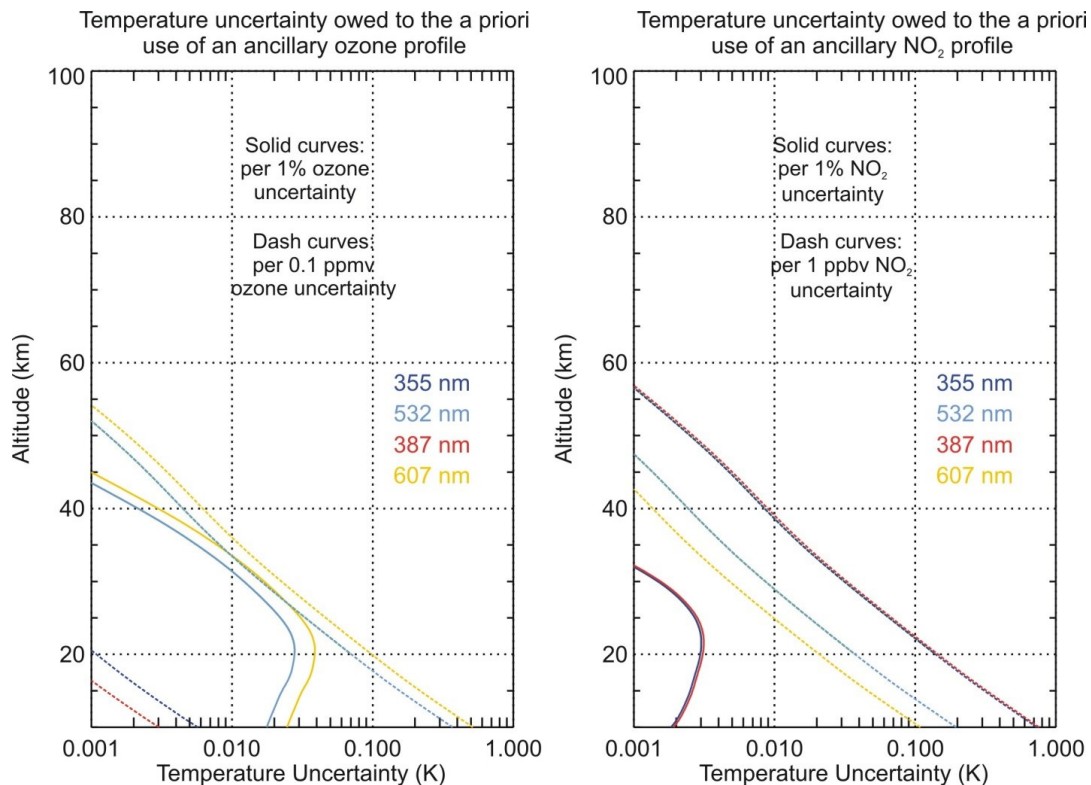

**Figure 7: Temperature uncertainty owed to the a priori use of ancillary ozone number density (left) and NO$_2$ number density (right) for the absorption correction. The results are shown per 1% uncertainty (solid curves), and for 1 ppmv ozone uncertainty (dash curves, left) and 1 ppbv NO$_2$ uncertainty (dash curves, right)**



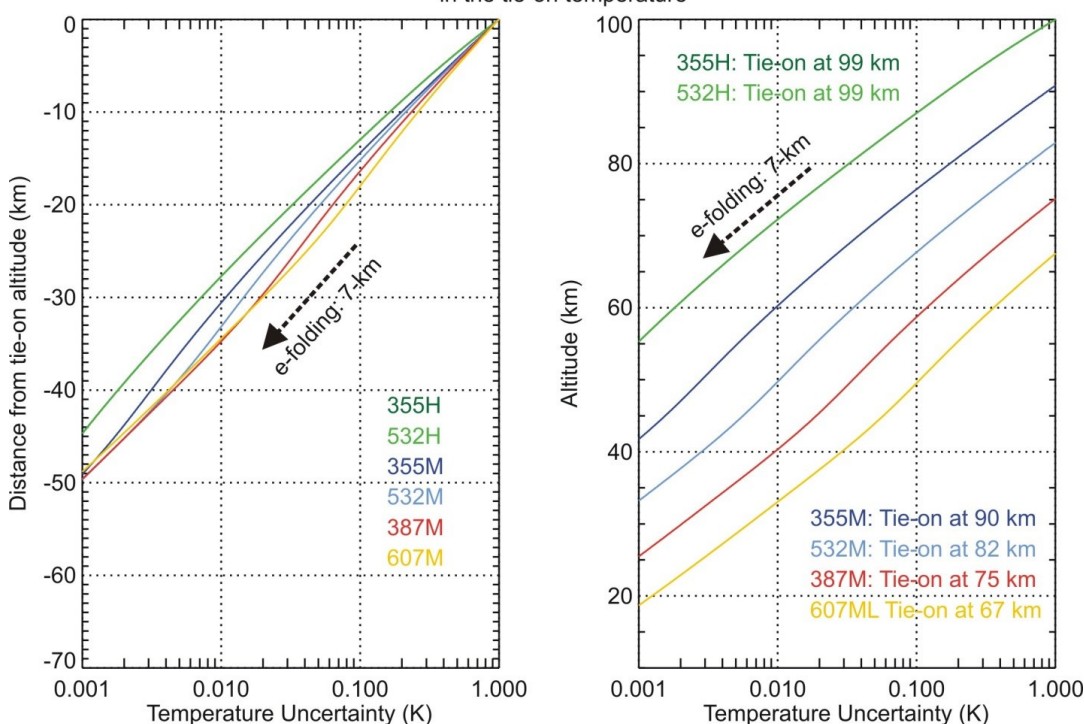

**Figure 8: Temperature uncertainty owed to a priori use of ancillary temperature to tie-on at the start of the density integration process. The results are shown per 1 K ancillary temperature uncertainty.**



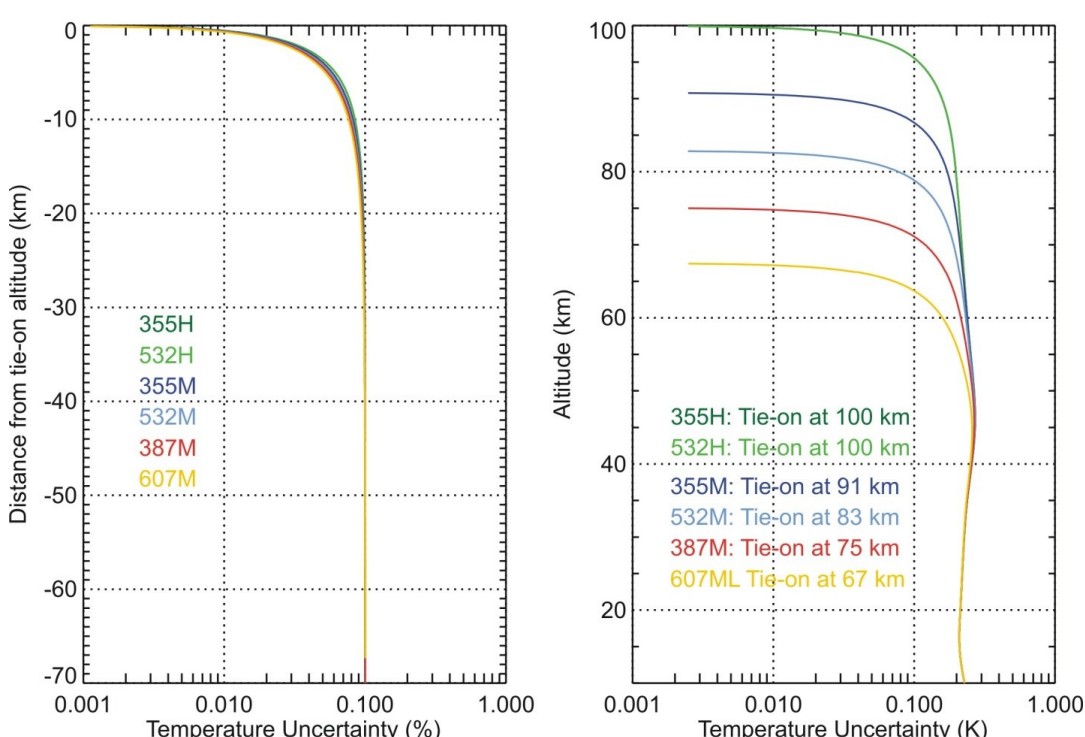

**Figure 9: Temperature uncertainty owed to 0.1% uncertainty in the acceleration of gravity. Left: Relative uncertainty (%) as a function of the distance from the tie-on altitude. Right: Absolute uncertainty (K) as a function of altitude**



Figure 10: Example of full uncertainty budget for the JPL-Mauna Loa Observatory temperature lidar, as computed using the expressions proposed in the present section (data taken during 2 hours on March 13, 2009)