# Peer review of "Proposed standardized definitions for vertical resolution and uncertainty in the NDACC lidar ozone and temperature algorithms. Part 3: Temperature uncertainty budget"

_Atmospheric Measurement Techniques, 2016_

## Referee Comment (RC1) · Anonymous Referee #1 · 1 Jun 2016

This manuscript presents a standardized definition for the computation of temperature uncertainty budget in NDACC Rayleigh temperature lidars. The main objective of these lidars is to provide high quality data for the long term monitoring of middle atmospheric temperature. This requires a very good characterization of the error budget and the possibility to intercompare the results obtained with the different instruments of the network. Up to now no effort has been made to homogenize the procedures applied for the determination of individual uncertainty components and their propagation in the data processing chain of lidar algorithms. This paper represents a very valuable contribution to the improvement of the data quality of NDACC temperature lidars. It will

be worth for publication in AMT after taking into account the remarks detailed below.

1) I suggest indicating since the beginning of section 3 the assumptions made in the measurement model. This will make the reading of the paper easier. For instance the assumption that the lidar is operated in photon counting mode appears only page 6, line 33and the assumption of a full overlap between the laser beam and the telescope field of view appears only page 7, line 25. It is implicitly assumed in the equations that the laser beam is fully vertical. It has to be explicitly said in the text.

2) Filtering process: It is indicated in the abstract that covariance terms must be taken into account when vertical filtering is applied but the propagation of uncertainties during the vertical filtering process is not presented in the text. I assume that it consists in a linear summation for systematic errors and quadratic summation for random errors but it has to be described for the full consistency of the paper.

3) Equation (18), page 6: when the vertical filtering is applied to the temperature, it has to be made on the inverse of temperature. This is due to the fact that density and temperature are anticorrelated at small vertical scales as shown in equation (13) where $N(k)$ appears in the denominator of the expression giving $T(k)$. If the filtering is applied to $T(k)$, Symmetrical fluctuations in $N(k)$ due to the random photon noise will induce asymmetric perturbations in $T(k)$ and a warm bias in the smoothed temperature profile.

4) Equation (23 for $U\_S(DET)(k)$ is not correct. It does not take into account the correlation between two consecutive terms $U\_(N ÌĚ(DET) )(k')$ and $U\_(N ÌĚ(DET)(k'+1)$ coming from equation (22). Both terms are depending on $UN(DET)(k'+1)$. Assuming that $N(k')$ and $N(k'+1)$ are about equal, the uncertainty in $US(DET)(k)$ is underestimated by a factor $\sqrt{2}$ by equation (23). This equation has to be corrected to take into account the correlation between two consecutive terms.

Typing errors: Page 19, line 11, some reference is missing as indicated by the message "Error! Reference source not found".

Page 22, line 8: ". . .. is compiled in Table." 2 is missing

Please also note the supplement to this comment:
http://www.atmos-meas-tech-discuss.net/amt-2016-122/amt-2016-122-RC1-supplement.pdf

———————————————

---

## Referee Comment (RC2) · Anonymous Referee #2 · 6 Jun 2016

**1   Overall**

The paper summarizes procedure and results for obtaining the uncertainty budget of atmospheric temperature profiles measured by laser radars using the weak light scattered back from molecules in the atmosphere. It is part of a series of similar papers summarizing results of an extensive report generated under support from the International Space Science Institute in Bern, Switzerland. It is good to have peer reviewed summaries of these findings in the scientific literature. The content of this paper is very well suited for Atmospheric Measurement Techniques. Although the paper does

not lead to a substantial revision of the main sources of uncertainty, which have been known for decades, it does provide a comprehensive framework. It explicitly addresses also the minor sources of error. Overall this is an important reference document. I recommend publication, after the following questions have been addressed.

**2   Major points**

- The **abstract** should give numbers (in K) for the typical uncertainty attributable to the different uncertainty sources at representative altitudes. A short summary of Fig. 10 would do. E.g. for a typical stratospheric / mesospheric NDACC lidar

  – overall uncertainty: from 15 to 50 km 0.1 to 1 K, from 60 to 80 km 1 to 10 K

  – signal detection uncertainty: from 15 to 50 km 0.1 to 1 K, from 60 to 80 km 1 to 10 K

  – far range tie-on temperature : 0.1 K at 60 km (30 km below tie-on altitude), 1 K at 75 km (15 km below tie-on, 10 K at 90 km (tie-on altitude)

  – background estimation uncertainty: < 0.01 to 0.1 K at most altitudes below 60 km, 0.1 to 1 K above 60 km

  – counter saturation correction uncertainty: < 0.01 to 0.1 K at all altitudes, if excessive count rates are avoided

  – uncertainty in correction for Rayleigh extinction: 1 K to < 0.1 K at altitudes from 10 to 20 km, negligible above 20 km

  – uncertainty due to other sources: < 0.1 K

- Outside of the abstract, it would be good to also have a table summarizing these various uncertainty values, for typical altitudes (say 30, 60, 80 km).

- I realize that the authors put a lot of work into estimating also the minor / almost negligible uncertainties. This is good, and needs to be done to show what is major and what is minor. However, from a user perspective, when reading the paper, especially, abstract, conclusions and looking at the final Figure 10, it would really help to bring out the major uncertainties much clearer, e.g. be marking them in bold, thicker lines, . . . Also please add some text that differentiates between major and minor uncertainties. Some of these uncertainties are really small and almost irrelevant. More guidance would really help users!!

- Certain other uncertainties are ignored completely: There is no mention of multiple scattering, which can invalidate Eq. 3. Yet, multiple scattering may be a root cause why so many intercomparisons between lidar measured temperatures and other sources (e.g. radiosondes) show a low bias up to 1 or 2 K for the lidar data (e.g. Reichardt and Reichardt, 2006). The authors should come back to this in conclusions and abstract, and should at least give some numbers/ ideas.

- Not dealing with additional aerosol scattering and extinction from the stratospheric aerosol layer is also a weak point for a paper with the scope of this manuscript. The authors should come back to this in conclusions and abstract, and should at least give some numbers/ ideas.

- Another weak point, in my opinion, is the omission of the effect of filter bandwidth on receiver efficiency altitude dependence, due to including / excluding rotational Raman wings for scattering from cold / warm regions of the atmosphere. (See e.g. She, 2001; Whiteman, 2003). This can also introduce temperature uncertainties up to 0.2 or 0.4 K, more relevant than some of the smaller uncertainties discussed here. Given the intended comprehensive scope of the manuscript, uncertainty / error sources of this magnitude should be at least mentioned in introduction, conclusions and abstract.

- Throughout the manuscript I find the term "background noise" confusing / misleading. Nearly always the authors mean "background" i.e. an underlying smooth curve, but not "noise" i.e. random deviations from a smooth curve. I suggest to replace "background noise" by "background" throughout this paper (and the two companion papers!!).

- While the paper extensively mentions uncertainties in Rayleigh-, ozone-, and nitrous oxide cross-sections, it never gives any values for the cross-sections themselves. Since this is intended as a reference publication, it would be very helpful for readers and for standardization, if the paper did include tables with recommended values for all of these cross-sections.

**3  Minor points**

pg.2, lines 10, 12: I suggest to replace "review" by "summarize" I don't think the authors are giving a proper independent review of their work here. Rather, they summarize what they found in their own report.

pg. 3, line 3: Should that not be "proposing" instead of "propose"?

pg. 4, line 27: I think the authors mean "transmission", not "thickness". Isn't optical thickness the quantity in the exponential function in Eq. 4, i.e. the $\tau$ in $T = \exp\left(-\tau\right)$, whereas T is the (one-way) transmission? Please check and correct.

pg. 7, lines 4, 5: Replace "noise" by "background"? See my major comment above. Also: Maybe (residual) fluorescence should be mentioned here too.

pg. 7, line 15: Please mention that dead-time correction in Eq. 10 is only a first order approximation. Generally, an exponential equation like Eq. 5 of Donovan et al. (1993) is needed.

[Figure]

pg. 8, line 15: Since this intends to be a reference publication: Please add a table giving (latitude dependent) recommended values for the constant $g_0, g_1, g_2$.

pg. 9, line 31: add "uncertainty in" before "the acceleration". Instead of "providing ... altitude dependent" maybe say "provided an altitude dependent formulation of gravity as in Eq. 15 is used".

pg. 11, line 3: Remove "a" before "Poisson". Maybe mention that Eq. 19 is only valid when $R(k)$ is the total number of counted photons in channel $k$.

pg. 14, line 21: Year / reference McGee is missing.

pg. 19, line 11: reference is missing.

pg. 21, last paragraph: I think it would be very helpful if the various uncertainty components introduced here were linked to the uncertainties used in NDACC lidar hdf files following the GEOMS hdf conventions of the Aura Validation Data Center. I think one or two paragraphs here, as well as some text in conclusions and abstract would be really warranted.

pg. 22, line 8: reference (to Table 2?) is missing.

pg. 22, last paragraph, also pg. 23, lines 30, 31: This may be right, but to me it is confusing. Are you not simply saying that averaging over many profiles will average out "random" uncertainties, but not "systematic uncertainties". Can this be reworded? Maybe show an equation that explains this? $\sqrt{a^2 + a^2} = \sqrt{2}a$ for uncorrelated uncertainties $a$, but $\sqrt{b^2 + 2bc + c^2} = b + c$ for correlated uncertainties $b, c$.

Figure 1: Too many lines. I think just showing one vertical sampling is enough. Lines have to be shifted for different signal strengths and vertical samplings anyways.

Figure 10: I think this is a very important Figure, because it shows the magnitudes of the different uncertainties. Unfortunately, in my copy I was not able to read the small print and figure out which line belongs to which uncertainty. Also, not all colored lines

appear in all plots. Please make the labels $u_{xxx}$ much larger, maybe omit the $u$ and just give the more important $xxx$. Please make sure to only show those $xxx$ that actually have lines in each panel. Many colors are very similar, and it is very hard to tell which line is which color. Please use different colors, or draw arrows from the labels to the lines.

**4 References**

Donovan, D.P., J.A. Whiteway, and A.I. Carswell, Correction for nonlinear photon-counting effects in lidar systems, Applied Optics, 32, 6742–6753, 1993.

Reichardt J., and S. Reichardt, Determination of cloud effective particle size from the multiple-scattering effect on lidar integration-method temperature measurements, Applied Optics, 45, 2796–2804, 2006.

She C.-Y., Spectral structure of laser light scattering revisited: Bandwidths of nonresonant scattering lidars, Applied Optics, 40, 4875–4884, 2001.

Whiteman, D.N., Examination of the traditional Raman lidar technique. I. Evaluating the temperature-dependent lidar equations, Applied Optics, 42, 2571–2592, 2003.

---

## Author Comment (AC1) · 6 Aug 2016

We would like to thank the reviewers for taking the time and making the effort to review our manuscript. Please find below our detailed responses to all points raised by the reviewers, which after being addressed, led to a greatly improved version of the manuscript.

Responses to Reviewer # 1:

1) Provide measurement model assumptions early in section 3: Following the reviewer's suggestion we added the following sentences at the beginning of section 3:

"We start with the most general form of the lidar equation (section 3.1), then we revert this equation (section 3.2) with the assumptions that 1) the beam is vertical, 2) there is complete overlap between the beam and the telescope field-of-view, and 3) detection mode is photon-counting only (section 3.3). The cases of analog detection and incomplete overlap are partially treated in the full ISSI Team Report (Leblanc et al., 2016a)." We also added the following sentence in section 3.3: "However, for some systems, analog signal counting statistics were reported to be consistent with a Poisson distribution (Whiteman et al., 2006), and therefore many aspects of the treatment of uncertainty owed to detection noise described in this manuscript is likely to apply to analog-to-digital converted signals." We added the following reference: "Whiteman, D. N., et al.: Raman Lidar Measurements during the International H2O Project. Part II: Case Studies, J. Atmos. Ocean Tech., 23, 170-183, 2006."

2) Filtering process: The reviewer is absolutely right: we have not included the propagation of uncertainty through the filtering process. All the details of this process are in the ISSI Team Report, and as of today, it is unclear why we did not include it in our "Part 3" manuscript (we have it in Part 2, Ozone DIAL). Nevertheless, we agree it was a mistake, and we have included it as a new section 4.11 (channel merging is now section 4.12). The reviewer is correct that for random components, we apply the quadratic sum of the filter coefficients-weighted uncertainties, and for fully correlated components, we apply a simple linear combination.

3) Filtering applied to T instead of 1/T Actually the fluctuations in N are not symmetrical because of the exponential decrease of N with altitude. As a result, the distribution of fluctuations around the mean is log-normal, not normal. When we detail the filtering processes we explicitly recommend smoothing the logarithm of the signal. On the other hand, the natural distribution of temperature fluctuations is normal, which is why we recommend to smooth T.

4) Correlation terms for Eq. 23 (old numbering): Yes it is correct that the correlation terms were neglected in this equation. We now use the full expression, with the

correlation coefficient included (now Eq. (26)). Theoretically speaking, this is an important correction. Following the reviewer's suggestion, we also included his suggested approximation, yielding the addition a Sqrt(2) factor in front of the original expression (now Eq. 27). However, quantitatively speaking, the impact on temperature uncertainty is negligible because the magnitude of this component is much smaller than that of the other two contributions in Eq. 28. We attached a supplemental PDF providing more details and two figures on this. Below is the modified text in our revised manuscript: "The detection noise uncertainty then needs to be propagated to the sum S defined in Eq. (14). This sum involves correlated terms as two consecutive terms contain two occurrences of the same values (k' and k'+1 first level, then k'+1 and k'+2 next level, etc.). The application of Eq. (2) to Eq. (14) in its mot general sense yields:

Eq. (26)

The correlation coefficients rk'k" between the terms and are not strictly known. However, with the realistic assumption that the values of two consecutive terms are almost equal (i.e., N values, g values and uN(DET) values), an approximation of Eq. (26) can be written:

Eq. (27)

This expression is different from an expression assuming that all terms are independent (it is a factor of larger), and it is also different from an expression assuming that all the terms are fully correlated (the weighed sum of all individual uncertainties). Though it differs from the theoretical expression, its magnitude once propagated to temperature is significantly smaller than the magnitude of the other terms contributing to temperature uncertainty owed to detection noise (see Eq. (28) below). For more accurate estimates of uS(DET), a full quantification of the correlation coefficients rk'k" is required. The value of those coefficients depends on the lidar signal magnitude, the lidar sampling resolution, and the amount of vertical smoothing applied. For vertically unsmoothed signals, a simple parameterization of altitude can be used, starting at the

value of 1 at the tie-on altitude, and decreasing exponentially to 0 several kilometers below. For vertically smoothed signals, the parametrization has to take into account the type of smoothing filter used and the number of filter coefficients as a function of altitude. The parameters of the correlation coefficients altitude-dependent function can be determined by running Monte-Carlo experiments assuming repeatable behavior of the actual lidar signals considered." We also added the following text right after Eq. 28: "The third term under the square root is much smaller than the first and second terms (typically by more than 1 order of magnitude). As a result, the inclusion or omission of the factor in Eq. (27) has almost no impact on the actual temperature uncertainty owed to detection noise"

Typos: Missing reference was "Figure 9", corrected. Table "2" corrected.

Responses to Reviewer # 2:

Major Comments:

Summary of quantitative estimates in abstract: Thank you for this suggestion, although it is difficult to provide actual values that are representative of a large number of instruments with so different specifications. Nevertheless we added the following paragraph to the abstract: "Using this standardized approach, an example of uncertainty budget is provided for the JPL lidar at Mauna Loa Observatory, Hawaii, which is typical of the NDACC temperature lidars transmitting at 355 nm. The combined temperature uncertainty ranges between 0.1 K and 1 K below 60 km with detection noise, saturation correction, and molecular extinction correction being the three dominant sources of uncertainty. Above 60 km and up to 10 km below the top of the profile, the total uncertainty increases exponentially from 1 K to 10 K due to the combined effect of random noise and temperature tie-on. In the top 10 km of the profile, the accuracy of the profile mainly depends on that of the tie-on temperature. All other uncertainty components remain below 0.1 K throughout the entire profile (15-90 km)" except the background noise correction uncertainty which peaks around 0.3-0.5 K. It should be kept in mind

that those quantitative estimates may be very different for other lidar instruments, depending on their altitude range and the wavelengths used."

Summary of quantitative estimates in table: Once again, it is difficult to provide actual values that are representative of a large number of instruments with so different specifications. However we added Table 3 (and some text to go with it) which somehow summarizes a typical uncertainty budget

Reviewer comment on identifying more clearly the more significant from the less significant components: This is a good point. We modified the text and the abstract to reflect that. Second paragraph of the abstract: "The identified uncertainty sources comprise major components such as signal detection, saturation correction, background noise extraction, temperature tie-on at the top of the profile, and absorption by ozone if working in the visible, as well as other components such as molecular extinction, the acceleration of gravity, and the molecular mass of air, whose magnitudes depend on the instrument, data processing algorithm, and altitude range of interest." Section 3.3: "The above input quantities are not listed in order of significance, but instead, in the order they are introduced into the lidar temperature model. Quantitatively, the most significant uncertainty components are typically detection noise (1) and temperature tie-on (10) at the top of the profile, and saturation correction (2) and molecular extinction (4 and 5) at the bottom of the profile." Conclusion: "In general, the largest uncertainty components include detection noise and temperature tie-on at the top of the profile, and saturation correction and molecular extinction at the bottom of the profile (see example in Table 3 and in Figure 10). Uncertainty contributions from absorption by NO2 and O3 are less significant, and the contributions from the acceleration of gravity and the molecular mass of air are negligible if those quantities are chosen accurately when introduced in the temperature retrieval."

Multiple scattering: It is correct that we had not mentioned multiple scattering. We have now added some comments on this issue in section 3.3, including the reference to the Reichardt and Reichardt, 2006 paper cited by the reviewer. Like for particulate extinction and backscatter, the effect of multiple scattering is difficult to quantify, and more importantly, even more difficult to standardize. The reference cited by the reviewer is a good example of work done on this topic, but literature, in general, lacks compilations of the fundamentals needed to quantify this effect unequivocally for all measurement conditions and instrumental configurations. We recognize that this issue is important and should be included among issues to be addressed in the near future, for example by forming another ISSI Team dedicated to the interference of particles in the ozone, temperature and water vapor lidar retrievals. We already had a sentence on this in sect. 3.3, but now we also have modified the last sentence of the conclusion to emphasize this need. This sentence now reads: "We therefore recommend the formation of new working groups, possibly in the form of ISSI Teams, whose tasks would be not only to extend the present work to the retrieval of other species, but also to propose a similar standardized treatment of uncertainty accounting for the interference by particulate backscatter and extinction, and multiple scattering". Finally we added a row in our new Table 3 to report a rough estimate of uncertainty associated with the interference by particles.

Aerosol scattering and extinction: Our response is very much an extension of our response to the multiple scattering issue. We modified the manuscript in a way that addresses the reviewer's concern on both multiple scattering and particulate backscatter and extinction.

Filter bandwidth: This particular effect has been quite scrutinized for water vapor Raman lidars because of the needs for some lidars to measure during the day, and therefore sue very narrow filters. Most NDACC backscatter temperature lidars are optimized for nighttime temperature measurements, and therefore typically use wide filters (0.8 nm or wider). For those widths, the impact of temperature is much smaller than the reviewer suggests. However, for the sake of completeness, we added some comments on this topic in section 3.3, specifically saying that an additional source of uncertainty should be introduced for narrow filters. Note that once again, recommending a standardized approach for this particular effect is very difficult, as the temperature dependence varies greatly with the actual position and width of the filter. The beginning of Sect. 3 has been modified with the following updated paragraph: "... 3) the lidar receiver uses wide-enough filters so that they are insensitive to the temperature dependence of the Raman spectrum, and 4)..." Sect. 3.3 has been modified with the following updated paragraph: "When the receiver field-of-view and the laser beam are known to not fully overlap, an additional "instrumentation-related" uncertainty component must be introduced to take into account the overlap correction (altitude-dependent term ïAÍ in Eq. (12)). Also, if the lidar receiver uses very narrow filters (typically narrower than 0.7 nm), another "instrumentation-related" uncertainty component must be introduced to take into account the temperature dependence of the Raman backscatter cross-sections (causing again the term ïAÍ in Eq. (12) to be altitude-dependent). Because the overlap function and the filter width and position are strongly instrument-dependent, a standardized approach for the treatment of those uncertainty components cannot be proposed here (beyond the scope of this paper). In the rest of this work, we will therefore assume full overlap and wide-enough filters to prevent an altitude dependence of the lidar transmission function"

Background noise: In fact, "noise" refers to whatever harms or dissimulates a "signal" (see for example the Merriam-Webster definition: unwanted electronic signals that harm the quality of something). This is the definition we have used throughout all 3 manuscripts, and for this reason, we have chosen to keep it as is in the manuscripts (it has been used in that sense before).

Recommended ozone and NO2 cross-sections: Indeed before submission of the original manuscript, the authors have lengthily discussed the inclusion or not of such technical, and somewhat subjective, information. We did include examples in Table 3 of our companion paper (Part 2), but we agreed that for more details, the reader should refer to the ISSI Team Report, where those things were investigated thoroughly (see Section 3.5 and Appendix D-E of the Report). In order to support the reviewer's suggestion,

we added a couple of sentences in the text that refer specifically to table 3 of our companion Part 2, and to the Section 3.5 and Appendix D and E of the ISSI Team Report. We added the following sentence in section 4.4 (Rayleigh scattering cross-sections): "A review of the different calculations and the associated uncertainties can be found in section 3.5 of the the ISSI team Report (Leblanc et al., 2016a)." We added the following sentence in section 4.6 (O3 absorption cross-sections): "For the ozone absorption cross-section, a review and assessment of the available datasets is summarized in section 3.5 and Appendix E of the ISSI team Report (Leblanc et al., 2016a)."

Minor comments:

All "minor" corrections suggested by the reviewer done except the following ones:

pg. 7, line 15: Please mention that dead-time correction in Eq. 10 is only a first order approximation. Generally, an exponential equation like Eq. 5 of Donovan et al. (1993) is needed Actually Eq. 11 (formerly Eq. 10) is an exact derivation originating in the theory of probability of a physical event to occur, as described by Muller 1973, assuming "zero-discriminator". The formula by Donovan represents a generalization of Muller formulation with a non-zero discriminator, and the first order solution of those generalized equation is Eq. 11. We preferred to keep the initial reference to Muller.

pg. 8, line 15: Since this intends to be a reference publication: Please add a table giving (latitude dependent) recommended values for the constant g0, g1, g2. Again, it is not our mandate here to prescribe specific values. However we added some text where we refer to the ISSI Team Report section 3.5 (and references therein) for the derivation of these coefficients

pg. 14, line 21: Year / reference McGee is missing Actually that was a mistake: It should have been Strauch, 1971, not McGee, 1993.

Figure 1, too many lines: We agree that the figure is busy. However, we do not think that it has reached a confusing point, and because of the amount of information it

contains, we think it is appropriate to maintain it with six different lidar/channel power configurations.

Please also note the supplement to this comment:
http://www.atmos-meas-tech-discuss.net/amt-2016-122/amt-2016-122-AC1-supplement.pdf

―――――――――――――――――

[Figure]

**Supplement:**

**Additional material addressing reviewer #1's comment on the correlation terms for Eq. 23 (old numbering, Eq. 26 new numbering):**

Yes it is correct that the correlation terms were neglected in this equation. We now use the full expression, with the correlation coefficient included (now Eq. (26)). Theoretically speaking, this is an important correction. Following the reviewer's suggestion, we also included his suggested approximation, yielding the addition a Sqrt(2) factor in front of the original expression (now Eq. 27).

We would like to add two other pieces of information:

**1) Parameterization of the correlation coefficients for use in Eq. 26:**
The most accurate numerical result for Eq. 26 is obtained if we parameterize the altitude-dependence of the correlation coefficients. Starting from a value of 1 at the tie-on altitude, this dependence typically decreases exponentially towards 0 as we integrate density downward. The scale-height of this decrease can be determined empirically by running Monte-Carlo experiments. An example is provided in figure 1 (next page). In this example, we ran 200 Monte Carlo simulations of the same atmospheric profile (forward model), where each simulation includes realistic random detection noise (Poisson statistics, all samples independent). A typical 355-nm Rayleigh high-intensity channel with temperature measurements between 30 and 80 km is used here.
In Figure 1, the black solid curve shows one of the 200 corrected signal profiles S (sum of g*N according to Eq. 14) as a function of altitude (with respect to the tie-on altitude). In this case the tie-on altitude was chosen at a STN ratio of 10, which is 15-20 km below the uppermost valid density point. The dotted red curve shows the standard deviation of S obtained from all 200 MC simulated profiles. The yellow dash-dotted curve shows the uncertainty $u_{S(DET)}$ obtained experimentally by computing all covariance terms using Eq. 26. The dash-dotted purple curve shows the uncertainty $u_{S(DET)}$ obtained if we assume full-correlation (r=1 for all z). The dotted blue curve shows the uncertainty $u_{S(DET)}$ obtained if we assume no correlation (r=0 for all z). The dash cyan curve shows the uncertainty $u_{S(DET)}$ obtained if we assume no correlation (r=0 for all z), but including the sqrt(2) factor in Eq. 27. Finally the green dash curve shows the uncertainty $u_{S(DET)}$ obtained by parametrizing the correlation coefficients as follows:

$$r_{k'k''}(k) = \exp\left( c_0 \frac{z(k) - z(k_{TTOP})}{H_0} \right) \qquad \text{with } k < k_{TTOP}, \ H_0 = 7 \text{ km, and } c_0 = 2.5 \qquad (R1)$$

**2) On the actual impact of $u_{S(DET)}$ (Eq. 26) on $u_{T(DET)}$ (Eq. 28):**
The actual "total" temperature uncertainty owed to detection noise, as expressed by Eq. 28, is formed of three terms under the square-root: the first and second terms, associated with the density uncertainty, are indeed much larger than the third term, this latter being the term propagated from Eq. 26). Therefore, the use of Eq. 27 with, or without sqrt(2) yields almost identical results. An example of the magnitudes of those terms are plotted in Figure 2 next page for a typical 355-High intensity Rayleigh channel.

[Figure]

Figure 1: parameterization of the correlation coefficients in Eq. 26

Corrected signal (Eq. 14) and detection noise uncertainty magnitude

- S defined by eq. (14)
- $u_{S(DET)}$ (eq. 26) if assuming full correlation
- $u_{S(DET)}$ (eq. 26) if assuming no correlation
- $u_{S(DET)}$ if approximated by (eq. 27)
- $\sigma_S$ obtained from MC simulations
- $u_{S(DET)}$ if computed from MC covariance
- $u_{S(DET)}$ if approximated by correlation coefficient model (eq. R1)

Figure 2: On the actual impact of $u_{S(DET)}$ (Eq. 26) on $U_{T(DET)}$ (Eq. 28)

- $u_{T(DET)}$ as defined by eq. (28)
- First term of $u_{T(DET)}$ under square-root in Eq. 28
- Second term of $u_{T(DET)}$ under square-root in Eq. 28
- Third term of $u_{S(DET)}$ under square-root assuming no correlation in Eq. 26
- Third term of $u_{S(DET)}$ under square-root assuming no correlation in Eq. 26, but with sqrt(2) included (eq. 27)

Temperature uncertianty owed to detection noise (K) (following Eq. 28)